# Consistency-Driven Calibration and Matching for Few-Shot Class Incremental Learning

**Qinzhe Wang**[1]*, **Zixuan Chen**[1]*, **Keke Huang**[1]†, **Xiu Su**[2], **Chunhua Yang**[1], **Chang Xu**[3]

[1] School of Automation, Central South University, China
[2] Big Data Institute, Central South University, China
[3] School of Computer Science, Faculty of Engineering, The University of Sydney, Australia

{wangqinzhe, chenzixuan, huangkeke, xiusu, ychh}@csu.edu.cn, c.xu@sydney.edu.au

## ABSTRACT

Few-Shot Class Incremental Learning (FSCIL) is crucial for adapting to the complex open-world environments. Contemporary prospective learning-based space construction methods struggle to balance old and new knowledge, as prototype bias and rigid structures limit the expressive capacity of the embedding space. Different from these strategies, we rethink the optimization dilemma from the perspective of feature-structure dual consistency, and propose a ***Consistency-driven Calibration and Matching (ConCM)*** framework that systematically mitigates the knowledge conflict inherent in FSCIL. Specifically, inspired by hippocampal associative memory, we design a *memory-aware prototype calibration* that extracts generalized semantic attributes from base classes and reintegrates them into novel classes to enhance the conceptual center consistency of features. Further, to consolidate memory associations, we propose *dynamic structure matching*, which adaptively aligns the calibrated features to a session-specific optimal manifold space, ensuring cross-session structure consistency. This process requires no class-number priors and is theoretically guaranteed to achieve geometric optimality and maximum matching. On large-scale FSCIL benchmarks including mini-ImageNet, CIFAR100 and CUB200, *ConCM* achieves state-of-the-art performance, with harmonic accuracy gains of up to 3.41% in incremental sessions. Code is available at: https://github.com/wire-wqz/ConCM

## 1 INTRODUCTION

Deep Neural Networks (DNNs) have achieved impressive success in various applications (He et al., 2016; Huang et al., 2017). Predominant approaches operate under closed-world assumptions, optimizing for static, predefined tasks. However, visual concepts in open-world environments are inherently unbounded and continuously evolving (McDonnell et al., 2023; Goodfellow et al., 2013). Moreover, acquiring sufficient training samples remains a critical bottleneck in DNNs training, especially for emerging categories (Wang et al., 2020). Humans, by comparison, exhibit lifelong learning capabilities: adapting new concepts from limited information while minimizing forgetting of past experiences. Inspired by this, FSCIL has emerged as a compelling paradigm that mimics human-like learning (Wang et al., 2023; Tao et al., 2020). FSCIL divides training into a base session with ample data and multiple incremental sessions with few-shot novel classes, requiring models to learn new concepts under limited supervision while retaining prior knowledge.

The scarcity of novel class samples induces overfitting in the backbone (Deng & Xiang, 2024). Thus, recent methods advocate fully training the backbone in the base session and freezing it thereafter, pairing it with a prototype-based classifier (Ayub & Fendley, 2022) for session-wise adaptation. While this decoupling strategy enhances stability, novel class features often lack adaptability to the embedding structure. This leads to knowledge conflict between classes, as shown in Figure 1 (a). Recent approaches adopt prospective learning to pre-allocate embedding space for novel classes in the base session and employ projection modules during incremental updates to ensure compatibility

---

*Equal contribution.
†Corresponding author.

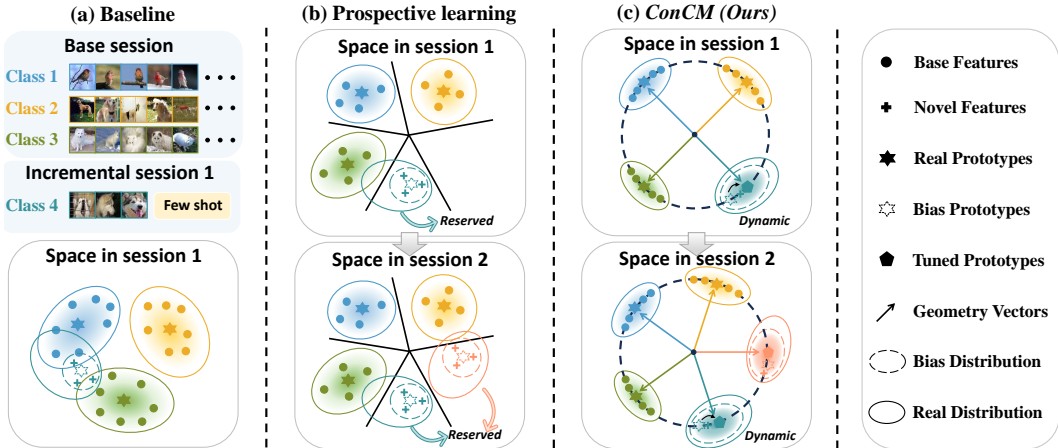

Figure 1: (a) **FSCIL setup and baseline.** It trains the backbone through the base session and freezes it. (b) **Prospective learning-based space construction method.** It pre-allocate fixed embedding spaces for novel classes, subject to the a priori assumptions of the class. (c) **The *ConCM* framework (Ours).** It leverages semantic attributes to calibrate novel class features by transferring knowledge from base classes, while incorporating a dynamical mechanism of feature-structure mathcing.

(Zhou et al., 2022a; Yang et al., 2023; Ahmed et al., 2024). However, by enhancing novel class performance at the expense of compressing old class embedding, these methods lack global structure optimization across sessions. As a result, the knowledge conflict remains unresolved.

In terms of embedding structure optimization, many methods enhance feature space generalization by designing specialized structures (Zhang et al., 2025; Ma'sum et al., 2025), as shown in Figure 1 (b). For instance, OrCo constructs orthogonal pseudo-targets to maximize inter-class distances (Ahmed et al., 2024), while NC-FSCIL pre-assigns classifier prototypes as an equiangular tight frame (ETF) to guide the model toward a fixed optimal configuration (Yang et al., 2023). Although these methods enhance structural adaptability, two key issues persist in FSCIL. First, the inherent feature bias in few-shot settings (Pan et al., 2024; Goswami et al., 2024) causes inconsistency between novel class prototypes and their true centers. Second, fixed structural constraints impose rigid priors on novel classes, restricting their matching flexibility and resulting in inconsistency between the actual and expected embedding structures. To this end, we explore a new unified perspective in FSCIL: *optimizing dual consistency between feature and structure through structured learning across incremental sessions to enable robust continual learning.*

The human ability to robust continual learning is largely attributed to hippocampal associative memory (Gutiérrez et al., 2024). Inspired by this, we propose a ***Consistency-Driven Calibration and Matching Framework (ConCM)***, which systematically mitigates the knowledge conflict inherent in FSCIL, as shown in Figure 1 (c). Specifically, to eliminate feature bias, we design *memory-aware prototype calibration*. It extracts generalized semantic attributes from base classes to build a class-related memory index, which is then queried and aggregated via meta-learning to calibrate novel class prototypes, thereby ensuring feature consistency. To consolidate memory associations, we further propose *dynamic structural matching* that constructs an evolving geometric structure and adaptively aligns features with it, thereby ensuring structural consistency across sessions. Theoretical analysis confirms that the proposed structure satisfies geometric optimality by equidistant prototype separation and achieves maximum matching by minimal global structural change.

In summary, our contributions are: (1) We alleviate the knowledge conflict in FSCIL from a unified perspective of consistency-structured learning. The proposed *ConCM* framework systematically resolves the dual consistency challenges of prototype bias and structure fixity. (2) To ensure consistency between novel class prototypes and their true centers, we draw inspiration from the hippocampal associative memory and propose memory-aware prototype calibration. (3) A geometric structure is constructed to jointly satisfy geometric optimality and maximum matching, along with a rigorous theoretical analysis. (4) Extensive evaluations on large-scale FSCIL benchmarks confirm the state-of-the-art performance, with ablation studies validating the contribution of each module.

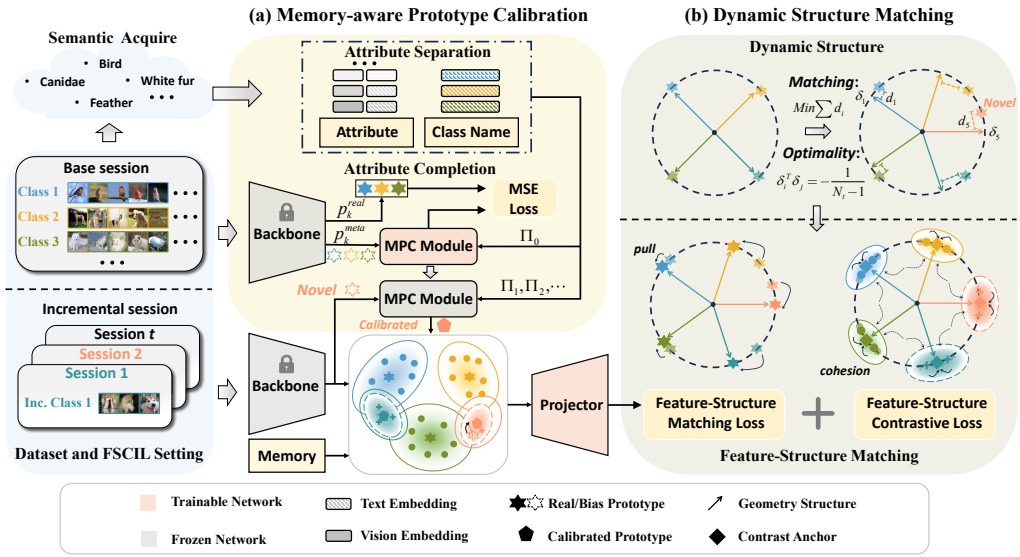

Figure 2: The proposed *ConCM* framework has two main modules: (a) MPC module. Generalized semantic attributes are extracted from base classes, *i.e.*, attribute separation, followed by meta learning–based retrieval and aggregation for prototype calibration, *i.e.*, attribute completion. (b) DSM module. Each incremental step dynamically updates embedding structure to ensure geometric optimality and maximum matching, while adaptively aligning features through loss-driven optimization.

## 2 RELATED WORKS

FSCIL aims to enable continual learning under limited supervision. We focus on optimization-based approaches, which mainly include: replay (Zhang et al., 2021; Hu et al., 2025) or distillation (Zhao et al., 2023; Kukleva et al., 2021), meta learning (Hersche et al., 2022; Zhou et al., 2022b; Chen et al., 2025), feature fusion-based (Goswami et al., 2024; Wang et al., 2023; Akyürek et al., 2022), and feature space-based methods (Zhou et al., 2022a; Yang et al., 2023; Ahmed et al., 2024). The latter two methods aim to obtain more efficient feature representations and are highly relevant to our method. Therefore, we focus on these methods below, with further details provided in Appendix B.

**Feature fusion-based FSCIL methods.** They aim to construct more accurate feature embeddings by integrating representations derived from diverse information sources or feature extraction methods. The semantic subspace regularization method (Akyürek et al., 2022) and TEEN (Wang et al., 2023) leverage semantic relations to fuse base and novel prototypes. PA (Liu et al., 2025) decouples common features within a family and transfers them to new species. In contrast, we construct memory-aware generalized semantic attributes to enhance semantic guidance.

**Feature space-based FSCIL methods.** Aiming to reserve embedding space for novel classes in prospective learning to mitigate knowledge conflict. FACT (Zhou et al., 2022a) creates virtual prototypes to reserve embedding space for novel classes. NC-FSCIL (Yang et al., 2023) pre-assigns fixed classifier as an equiangular tight frame and uses dot-regression loss to maintain feature-classifier alignment. OrCo (Ahmed et al., 2024) constructs a globally orthogonal embedding space, combining feature perturbation and contrastive learning to reduce inter-class interference. In contrast, we construct an evolving geometric structure to ensure consistency across sessions.

## 3 THE PROPOSED FRAMEWORK: *ConCM*

**Overview of the framework.** In Section 3.1, we detail the FSCIL task and analyze its core challenges by preliminary experiments: the feature-structure dual consistency problem. We then propose a *ConCM* framework to mitigate the conflict between learning and forgetting, depicted in Figure 2. For feature inconsistency, we propose memory-aware prototype calibration (MPC in Section 3.2). For structure inconsistency, we propose dynamic structure matching (DSM in Section 3.3).

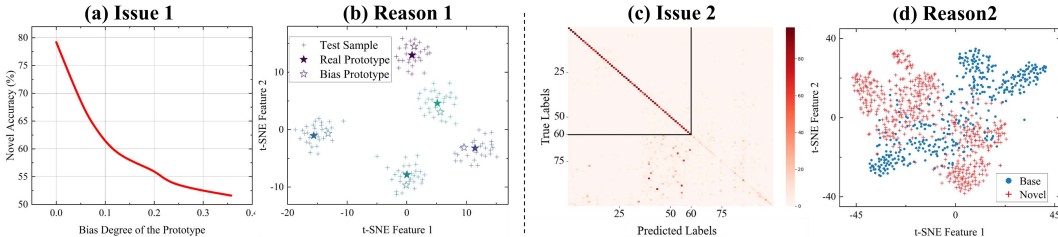

Figure 3: Preliminary results.We identify two issues and causes: **(a) Issue 1:** The accuracy on novel classes consistently declines as prototype deviation increases, caused by **(b) Reason 1:** Novel class prototypes deviate from the true centers, *i.e.*, *feature inconsistency*. **(c) Issue 2:** false positive classification, caused by **(d) Reason 2:** feature embedding confusion, *i.e.*, *structure inconsistency*.

### 3.1 TASK REDEFINITION AND PROBLEM ANALYSIS

**FSCIL task redefinition.** As shown in Figure 1, FSCIL consists of a base session $\mathcal{D}^0$ and multiple incremental sessions $\mathcal{D}^1, \cdots, \mathcal{D}^T$. Each session $\mathcal{D}^t$ is represented as a triplet of training images, one-hot encoded labels, and textual class names, *i.e.*, $\mathcal{D}^t = \{(x_i, y_i, c_i)\}_{i=1}^{|\mathcal{D}^t|}$. In the base session, we train the backbone on a dataset with sufficient base class samples and freeze it as a feature extractor. A projector $g(\cdot; \theta_g) \in \mathbb{R}^{d_g}$ is further trained to map features into a geometric space, enabling initial alignment with the target structure $\Delta_t = [\delta_1, \delta_2, \cdots, \delta_{N_t}]$. In the following $T$ incremental sessions (where $T$ is unknown), the model has access only to $\mathcal{D}^t$. Using $\mathcal{D}^t$, we fine-tune the projector $g(\cdot; \theta_g)$ to expand the geometric space and adaptively align features to novel classes $\mathcal{C}^t$ while preserving base classes $\mathcal{C}^0$ performance. Each incremental session is formulated as a typical $N$-way $K$-shot few-shot learning task, *i.e.*, $|\mathcal{D}^t| = NK$. Label spaces do not contain any overlap $\mathcal{C}^t \cap \mathcal{C}^{t'} = \emptyset, \forall t \neq t'$. Notably, before projection we add a prototype calibration network $h(\cdot; \theta_h) \in \mathbb{R}^{d_f}$ that leverages textual class name $c_i$ to extract latent semantic attributes and calibrate novel class prototypes.

After training, session $t$ is evaluated using a nearest class mean (NCM) classifier, which assigns class labels based on the distance between projected feature vectors and corresponding geometric vectors. The test set $\tilde{\mathcal{D}}^t$ spans all seen classes, implying $\tilde{\mathcal{C}}^t = \cup_{i=0}^{t} \mathcal{C}^i$, with $N_t = \left|\tilde{\mathcal{C}}^t\right|$ total classes.

**Consistency problems in FSCIL.** Preliminary experiments on mini-ImageNet reveal two critical issues: (1) *Feature Inconsistency.* The accuracy of novel classes declines as prototype bias (it is quantified as $1 - cos(p, p_{real})$) increases (Figure 3 (a)) caused by the misalignment between few-shot prototypes and actual class centers (Figure 3 (b)). This motivates us to calibrate prototypes to ensure consistency with the true centers(Section 3.2). (2) *Structure Inconsistency.* Despite ensuring prototype consistency by calibration, novel samples are often misclassified as old classes (Figure 3 (c)). The root cause lies in the implicit constraints imposed by the fixed space on novel classes, which hinders effective optimization and causes structure inconsistency, ultimately leading to inter-class confusion of projector output features (Figure 3 (d)). Accordingly, we propose to dynamically refine the geometric structure to maintain cross-session consistency (Section 3.3).

### 3.2 CONSISTENCY-DRIVEN MEMORY-AWARE PROTOTYPE CALIBRATION

Hippocampal associative memory, essential for human continual learning, operates in two stages (Gutiérrez et al., 2024; Luo et al., 2022): first, perceptual information is encoded and separated into high-level representations, constructing a memory index by neural routing; second, when partial perceptual signals are received, the associated memory in the index is retrieved and integrated to reconstruct a complete representation. Thus, we mimic human associative memory with **attribute separation** and **attribute completion** to calibrate novel class prototypes, defining attributes as latent semantic attributes that are class-discriminative and environment-invariant (Zhang et al., 2023).

**Attribute separation.** Base classes, with abundant and diverse samples, provide text labels that embed rich semantic information. Therefore, we first leverage WordNet (Ge et al., 2022) to parse base class text labels $\{c_i\}_{i=1}^{|\mathcal{C}^0|}$ and obtain semantic extensions such as synonyms and hypernyms, from which latent semantic attributes are extracted. The words extended from class labels are further

integrated into a candidate attribute pool $\mathcal{A} = \{a_i\}_{i=1}^{N_a}$ and encoded as word embeddings $\mathcal{S}_a = \{s_{a_i}\}_{i=1}^{N_a}$. In addition, we construct a visual embedding prototype $f_{a_i}$ for each attribute, defined as the mean feature of all samples possessing that attribute.

$$f_{a_i} = \frac{1}{|\mathcal{D}_{a_i}^0|} \sum_{(x,y) \in \mathcal{D}_{a_i}^0} f(x, \theta_f); \quad \mathcal{F}_a = \{f_{a_i}\}_{i=1}^{N_a} \tag{1}$$

The set of attribute visual prototypes is denoted as $\mathcal{F}_a$. Extending to any session $t$, we similarly encode the current class text labels $\{c_i\}_{i=1}^{|\mathcal{C}^t|}$ into word embeddings $\mathcal{S}_c^t = \{s_k\}_{k=1}^{|\mathcal{C}^t|}$ for attribute completion. A binary association matrix $\mathcal{R}^t = [r_1, r_2, \cdots, r_{|\mathcal{C}^t|}] \in \mathbb{R}^{N_a \times |\mathcal{C}^t|}$ is further introduced to match current classes with attributes in pool, where $r_{a_i k} = 1$ if class $k$ has attribute $a_i$, and $r_{a_i k} = 0$ otherwise. Therefore, through attribute separation, the semantic knowledge set of session $t$ is represented as $\Pi_t = \{\mathcal{S}_a, \mathcal{F}_a, \mathcal{S}_c^t, \mathcal{R}^t\}$.

**Attribute completion.** As the core of attribute completion, the MPC network adopts an encode–aggregate–decode architecture. The encoder estimates the low-dimensional representation of both attribute visual prototype $\mathcal{F}_a$ and class prototype $p_k$. The aggregator performs attribute retrieval based on relevance weight $w_{a_i}^k$. The decoder outputs the calibrated prototype $\hat{p}_k$. $w_{a_i}^k$ is derived by cross-attention (Vaswani et al., 2017). Specifically, Semantic association are evaluated by the similarity between word embeddings of the base attribute pool $\mathcal{S}_a$ and those of the current class labels $\mathcal{S}_c^t$, while visual association are measured by the distance between visual prototypes $\mathcal{F}_a$ and $p_k$. Therefore, by jointly considering these two associations, we have

$$\xi_k = h_e(p_k; \theta_h^e) + \sum_{i=1}^{N_a} Softmax\left(w_{a_i}^k\right) \times h_e(f_{a_i}; \theta_h^e); \quad \hat{p}_k = h_d(\xi_k; \theta_h^d) \tag{2}$$

$$w_{a_i}^k = \left( \frac{\left\langle h_{g_1^a}(s_{a_i}, \theta_h^{g_1^a}), h_{g_1^c}(s_k, \theta_h^{g_1^c}) \right\rangle}{2\sqrt{d_s}} + \frac{\left\langle h_{g_2^a}(f_{a_i}, \theta_h^{g_2^a}), h_{g_2^c}(p_k, \theta_h^{g_2^c}) \right\rangle}{2\sqrt{d_f}} \right) \times r_{a_i,k} \tag{3}$$

where $w_{a_i}^k$ represent the relevance weight of attribute $a_i$ to class $k$, and $\xi_k$ represents the aggregate output. $\theta_h^e$ and $\theta_h^d$ are the encoder and decoder parameters. $\theta_h^g = \left\{ \theta_h^{g_1^a}, \theta_h^{g_2^a}, \theta_h^{g_1^c}, \theta_h^{g_2^c} \right\}$ denotes the parameters of the attention layer. $d_s$ and $d_f$ are the dimensions of semantic and visual embeddings. $r_{a_i k}$ serves to mask according to the binary class–attribute associations. $\langle \cdot, \cdot \rangle$ represents the vector inner product. In summary, all parameters of the network are represented by $\theta_h = \left\{ \theta_h^e, \theta_h^d, \theta_h^g \right\}$.

The network parameters are trained via meta-learning (Snell et al., 2017), where a series of $K$-shot episodic tasks are constructed in the base session to obtain biased prototypes $p_k^{meta}$, and the actual base class prototypes $p_k^{base}$ are used as supervision. The network is trained with an MSE loss to learn completing associated attributes.

$$\mathcal{L}_{MSE}(p_k^{meta}) = MSE(h(p_k^{meta}, \Pi_0; \theta_h), p_k^{base}), \ 0 < k \leq N_0 \tag{4}$$

After training, we similarly extend to session $t$, where the completed prototypes $\hat{p}_k = h(p_k, \Pi_t; \theta_h)$ are obtained and combined by weighting to form the final class prototypes:

$$\hat{p}'_k = \alpha \times p_k + (1 - \alpha) \times \hat{p}_k, \ \hat{\mathcal{D}}_k^t \sim Aug(\hat{p}'_k), \ N_{t-1} < k \leq N_t \tag{5}$$

where $\alpha$ controls the calibration strength. In addition, since old class samples are inaccessible and novel class samples are insufficient, we further construct an augmented dataset $\hat{\mathcal{D}}_k^t$ via Gaussian sampling for subsequent projector training. More details are provided in the Appendix D.

### 3.3 CONSISTENCY-DRIVEN DYNAMIC STRUCTURE MATCHING

**Dynamic structure.** Structure inconsistency in feature space is a key source of knowledge conflict. Our goal is to optimize the projector to construct a dynamic structure that satisfies both geometric optimality and maximum matching. These two properties are defined as follows.

***Definition1 (Geometric Optimality):*** The neural collapse theory reveals an optimal feature structure formed (Papyan et al., 2020). A structural matrix $\Delta_t = [\delta_1, \delta_2, \cdots, \delta_{N_t}] \in \mathbb{R}^{d_g \times N_t} (d_g > N_t)$

satisfies geometric optimality if the following condition holds.

$$\forall i, j, \ \delta_i{}^\top \delta_j = \frac{N_t}{N_t - 1} \lambda_{i,j} - \frac{1}{N_t - 1} \tag{6}$$

where $\lambda_{i,j} = 1$ when $i = j$, and 0 otherwise. It implies that prototypes are equidistantly separated.

***Definition2 (Maximum Matching):*** Fixed structures constrain matching for novel classes. Therefore, we aim to embed new classes with minimal structural change, which can be expressed as:

$$\Delta_t = \underset{\delta_i \in \Delta_t}{\arg\max} \sum_{i=1}^{N_t} \langle \delta'_i, \delta_i \rangle \tag{7}$$

We maximize the similarity between the optimal target structure $\Delta_t$ and the initial structure $\Delta'_t = \left[ \delta'_1, \delta'_2, \cdots, \delta'_{N_t} \right] \in \mathbb{R}^{d_g \times N_t}$ (The historical structure that incorporates the embedding of novel classes) to distill structural knowledge and ensure minimal structural change.

Base class prototypes and covariance diagonal are stored. Each session generates training data $Aug(\Omega_t)$ through Gaussian sampling-based prototype augmentation from the prototype repository $\Omega_t = \{p_b^{base}\}_{b=1}^{N_0} \cup \{\hat{p}'_k\}_{k=N_0+1}^{N_t}$. The projected mean serves as the initial structure $\Delta'_t$. ***Theorem 1*** describes the structure optimization.

***Theorem1:*** For session $t$, the dynamic structure that satisfies geometric optimality and maximum matching can be updated based on the following formulation.

$$W \Lambda V^\top = \Delta'_t \left( I_{N_t} - \frac{1}{N_t} \mathbf{1}_{N_t} \mathbf{1}_{N_t}^\top \right), \quad U_t = W V^\top \tag{8}$$

$$\Delta_t = \sqrt{\frac{N_t}{N_t - 1}} U_t \left( I_{N_t} - \frac{1}{N_t} \mathbf{1}_{N_t} \mathbf{1}_{N_t}^\top \right) \tag{9}$$

where $U_t = [u_1, u_2, \cdots u_{N_t}] \in \mathbb{R}^{d_g \times N_t}$ is the column-wise orthogonal matrix, $I_{N_t} \in \mathbb{R}^{N_t \times N_t}$ is the identity matrix, and $\mathbf{1}_{N_t} \in \mathbb{R}^{N_t \times 1}$ is an all-ones column vector. $W \in \mathbb{R}^{d_g \times N_t}$, $\Lambda \in \mathbb{R}^{N_t \times N_t}$ and $V^\top \in \mathbb{R}^{N_t \times N_t}$ represents the compact SVD result.

***Proof:*** for any pair of vectors $\delta_i$ and $\delta_j$, we have:

$$\forall i, j, \ \delta_i{}^\top \delta_j = \frac{N_t}{N_t - 1} \cdot (u_i{}^\top u_j - \frac{1}{N_t}) \tag{10}$$

According to the properties of singular value decomposition, $W$ is a column-wise orthogonal matrix, *i.e.*, $W^\top W = I_{N_t}$ and $V$ is an unitary matrix, *i.e.*, $V^\top V = V V^\top = I_{N_t}$. It can be deduced that $U_t$ is also a column-wise orthogonal matrix, *i.e.*, $U_t^\top U_t = I_{N_t}$. Therefore, we satisfy geometric optimality, *i.e.*, Eq. (6). Furthermore, Eq. (7) can be expressed as:

$$U_t = \underset{U_t^\top U_t = I_{N_t}}{\arg\max} \ tr(\Delta'_t{}^\top \sqrt{\frac{N_t}{N_t - 1}} U_t (I_{N_t} - \frac{1}{N_t} \mathbf{1}_{N_t} \mathbf{1}_{N_t}^\top)) \tag{11}$$

Let $k = \sqrt{\frac{N_t}{N_t - 1}}$, $M = I_{N_t} - \frac{1}{N_t} \mathbf{1}_{N_t} \mathbf{1}_{N_t}^\top$, where $M$ is a symmetric matrix. Hence, we obtain:

$$U_t = \underset{U_t^\top U_t = I_{N_t}}{\arg\max} \ k \cdot tr((\Delta'_t M)^\top U_t) \tag{12}$$

This is a classic orthogonally constrained trace maximization problem. Let $W \Lambda V^\top = \Delta'_t M$ denote the SVD matrix. The optimal solution under the optimal matching relationship is given by $U_t = W V^\top$. Please refer to Appendix E for more details.

**Feature-structure matching.** $Aug(\Omega_t)$ is taken as input, and the projector $g(\ ; \theta_g)$ is optimized to match the target structure $\Delta_t$ by the joint feature-structure matching loss and contrast loss. The matching loss between projected vector $z_i$ and its corresponding structure vector $\delta_k$ is defined as:

$$\mathcal{L}_{Match}(z_i) = -\log \frac{\exp\left(\langle z_i, \delta_k \rangle\right)}{\sum_{j=1}^{N_t} \exp\left(\langle z_i, \delta_j \rangle\right)} \tag{13}$$

This classification loss minimizes the distance between projected class means and their corresponding $\delta_k$. We also employ Supervised Contrastive Loss (SCL)(Chen et al., 2020) to enhance intra-class

Table 1: **SOTA comparison on mini-ImageNet. AHM** denotes the average harmonic mean. **FA** denotes the Top-1 accuracy in final session. The top two rows list CIL and FSL results implemented in the FSCIL setting. Detailed results for the remaining datasets are presented in the Appendix.

| Methods | Base Acc | Session-wise Harmonic Mean (%) ↑ | | | | | | | | AHM | FA |
|---|---|---|---|---|---|---|---|---|---|---|---|
| | | 1 | 2 | 3 | 4 | 5 | 6 | 7 | 8 | | |
| iCaRL(Rebuffi et al., 2017) | 61.31 | 8.45 | 13.86 | 14.92 | 13.00 | 14.06 | 12.74 | 12.16 | 11.71 | 12.61 | 17.21 |
| IW(Qi et al., 2018) | 61.78 | 25.32 | 20.45 | 22.62 | 25.48 | 22.54 | 20.66 | 21.27 | 22.27 | 22.58 | 41.26 |
| FACT(Zhou et al., 2022a) | 75.78 | 27.20 | 27.84 | 27.94 | 25.17 | 22.46 | 20.54 | 20.88 | 21.25 | 24.16 | 48.99 |
| CEC(Zhang et al., 2021) | 72.17 | 31.91 | 31.84 | 30.98 | 30.74 | 28.14 | 26.78 | 26.96 | 27.42 | 29.35 | 57.95 |
| C-FSCIL(Hersche et al., 2022) | 76.60 | 9.74 | 20.53 | 28.68 | 31.93 | 34.85 | 35.05 | 37.72 | 37.92 | 29.55 | 51.09 |
| LIMIT(Zhou et al., 2022b) | 73.27 | 40.34 | 33.58 | 31.81 | 31.74 | 29.32 | 29.11 | 29.57 | 30.28 | 31.97 | 48.08 |
| MetaNC(Ran et al., 2024) | 79.05 | 30.33 | 30.55 | 38.25 | 29.44 | 21.34 | 28.85 | 41.92 | 42.09 | 32.85 | 53.11 |
| SAVC(Song et al., 2023) | 80.02 | 42.40 | 37.80 | 36.56 | 37.20 | 33.53 | 31.72 | 31.98 | 32.94 | 35.52 | 54.34 |
| LCwoF(Kukleva et al., 2021) | 64.45 | 41.24 | 38.96 | 39.08 | 38.67 | 36.75 | 35.47 | 34.71 | 35.02 | 37.49 | 37.69 |
| FACL(Nema & Kurmi, 2025) | 86.23 | 50.21 | 40.35 | 37.45 | 39.59 | 35.61 | 32.32 | 32.66 | 34.22 | 37.80 | 58.63 |
| CLOSER(Oh et al., 2024) | 76.97 | 43.58 | 40.36 | 40.41 | 39.35 | 37.76 | 35.72 | 36.10 | 37.58 | 38.86 | 53.61 |
| BiDist(Zhao et al., 2023) | 74.67 | 42.42 | 43.86 | 43.87 | 40.34 | 38.97 | 38.01 | 36.85 | 38.47 | 40.35 | 55.69 |
| ADBS(Li et al., 2025) | 79.53 | 44.88 | 45.60 | 39.00 | 42.61 | 41.68 | 40.58 | 42.93 | 43.12 | 42.58 | 55.24 |
| TEEN(Wang et al., 2023) | 74.70 | 52.39 | 47.25 | 44.53 | 43.54 | 41.27 | 39.97 | 39.96 | 40.42 | 43.66 | 51.80 |
| Mamba-FSCIL(Li et al., 2024) | 84.93 | 58.01 | 53.33 | 49.98 | 48.98 | 44.06 | 40.14 | 41.73 | 41.72 | 47.25 | 59.36 |
| NC-FSCIL(Yang et al., 2023) | 84.07 | 62.34 | 61.04 | 55.93 | 53.13 | 49.68 | 47.08 | 46.22 | 45.57 | 52.62 | 57.97 |
| OrCo(Ahmed et al., 2024) | 83.30 | 67.14 | 64.12 | 60.71 | 57.16 | 55.34 | 51.52 | 50.96 | 51.44 | 57.30 | 56.04 |
| **ConCM (Ours)** | 83.97 | **70.34** | **66.59** | **63.38** | **59.59** | **57.05** | **53.95** | **53.49** | **53.92** | **59.78** | **59.92** |
| *Better than second* | | +3.20 | +2.47 | +2.67 | +2.43 | +1.71 | +2.43 | +2.53 | +2.48 | +2.48 | +0.56 |

compactness. By treating the structure vector as an anchor and including it in the positive set, the model learns structural information explicitly.

$$\mathcal{L}_{Cont}(z_i) = -\frac{1}{|P_i|} \sum_{z_j \in P_i} \frac{\exp(\langle z_i, z_j \rangle / \tau)}{\sum_{z_k \in N_i} \exp(\langle z_i, z_k \rangle / \tau)} \quad (14)$$

$P_i$ represents the positive sample set, which includes the anchor and instances from the same class. The negative sample set $N_i$ includes instances from other categories. $\tau$ is the temperature parameter. Therefore, combining Eq. (13) and Eq. (14) yields the final loss for optimizing the projector.

$$\mathcal{L}_{Proj} = \mathcal{L}_{Match} + \mathcal{L}_{Cont} \quad (15)$$

## 4 EXPERIMENTS

In Section 4.1, the main experiment details are introduced. In Section 4.2, a comparison is made with SOTA methods on mainstream benchmarks. Section 4.3 provides a comprehensive analysis to demonstrate the superiority of the proposed method and the effectiveness of each module.

### 4.1 MAIN EXPERIMENTAL DETAILS OF FSCIL

We evaluated the *ConCM* on three FSCIL benchmark: mini-ImageNet(Deng et al., 2009), CI-FAR100 (Krizhevsky et al., 2009) and CUB200 (Wah et al., 2011). The dataset split followed Zhang et al. (2021). For a fair comparison with existing methods (Ahmed et al., 2024; Zhao et al., 2023; Yang et al., 2023), we replay 5 samples per incremental class, consistent with prior work. All expeiments are conducted on a RTX 4090 GPU. Please refer to Appendix F for more details.

### 4.2 COMPARISON TO STATE-OF-THE-ART

In this section, we compare the proposed *ConCM* with the latest SOTA methods. The experimental results are provided in Table 1, Table 2 and Figure 4. We adopt the Harmonic Mean (HM) (Ahmed et al., 2024) as a balanced metric. The proposed method outperforms previous SOTA methods across all datasets, with the highest performance improvement in incremental sessions of 3.20% on

Table 2: **SOTA comparison on CIFAR100. AHM** denotes the average harmonic mean. **FA** denotes the Top-1 accuracy in final session.

| Methods | Base Acc | Session-wise Harmonic Mean (%) ↑ | | | | | | | | AHM | FA |
|---|---|---|---|---|---|---|---|---|---|---|---|
| | | 1 | 2 | 3 | 4 | 5 | 6 | 7 | 8 | | |
| FACT(Zhou et al., 2022a) | 78.32 | 42.85 | 38.21 | 33.03 | 32.35 | 31.71 | 34.15 | 33.37 | 33.62 | 34.91 | 51.83 |
| CEC(Zhang et al., 2021) | 79.63 | 41.98 | 39.18 | 34.25 | 33.27 | 33.84 | 33.49 | 32.90 | 32.16 | 34.45 | 49.08 |
| C-FSCIL(Hersche et al., 2022) | 77.35 | 31.47 | 28.33 | 26.68 | 23.11 | 24.85 | 24.29 | 23.04 | 23.49 | 25.66 | 49.32 |
| LIMIT(Zhou et al., 2022b) | 72.93 | 39.98 | 36.98 | 33.24 | 32.64 | 32.28 | 32.66 | 31.96 | 31.40 | 33.89 | 50.84 |
| MetaNC(Ran et al., 2024) | 79.17 | 31.13 | 25.75 | 31.60 | 22.87 | 18.34 | 19.26 | 28.40 | 27.04 | 25.55 | 51.99 |
| SAVC(Song et al., 2023) | 78.07 | 38.69 | 34.95 | 30.42 | 28.90 | 30.30 | 31.59 | 31.68 | 30.79 | 32.16 | 51.93 |
| FACL(Nema & Kurmi, 2025) | 83.65 | 34.79 | 33.90 | 30.12 | 30.09 | 29.31 | 29.24 | 31.38 | 30.17 | 31.13 | 56.74 |
| CLOSER(Oh et al., 2024) | 75.95 | 48.31 | 44.98 | 41.99 | 40.87 | 40.02 | 41.80 | 41.64 | 40.23 | 42.48 | 53.55 |
| BiDist(Zhao et al., 2023) | 78.72 | 47.87 | 42.68 | 40.18 | 37.02 | 35.42 | 33.80 | 33.25 | 30.94 | 37.65 | 46.84 |
| ADBS(Li et al., 2025) | 80.68 | 38.80 | 34.39 | 32.84 | 32.82 | 33.25 | 32.62 | 29.75 | 29.33 | 32.98 | 48.50 |
| TEEN(Wang et al., 2023) | 78.47 | 46.66 | 40.28 | 36.88 | 35.32 | 35.55 | 36.28 | 35.43 | 34.97 | 37.67 | 51.62 |
| Mamaba-FSCIL(Li et al., 2024) | 82.80 | 42.18 | 47.29 | 45.10 | 42.86 | 43.05 | 42.14 | 42.37 | 40.77 | 43.22 | 57.51 |
| NC-FSCIL(Yang et al., 2023) | 82.52 | 56.66 | 54.41 | 49.70 | 45.27 | 44.36 | 46.65 | 44.22 | 41.87 | 47.89 | 56.11 |
| OrCo(Ahmed et al., 2024) | 80.08 | 72.11 | 63.92 | 56.90 | 55.23 | 53.38 | 54.03 | 51.78 | 49.57 | 57.12 | 52.19 |
| **ConCM (Ours)** | 82.82 | **72.27** | **67.33** | **60.09** | **57.08** | **54.93** | **55.21** | **52.95** | **52.51** | **59.05** | **58.33** |
| *Better than second* | | +0.16 | **+3.41** | +3.19 | +1.85 | +1.55 | +1.18 | +1.17 | +2.94 | +1.97 | +0.82 |

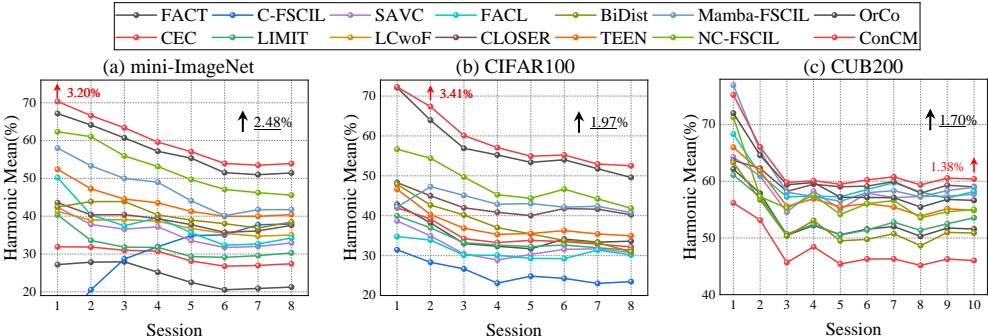

Figure 4: **SOTA comparisons on three FSCIL benchmark datasets.** Performance curve is harmonic mean accuracy. The underlined denotes the average performance improvement. The red denotes the highest performance improvement.

mini-ImageNet, 3.41% on CIFAR100, and 1.70% on CUB200. Compared to related works, NC-FSCIL(Yang et al., 2023) is a static structure optimization method without considering the inherent bias of the embedded features. ConCM achieves structural dynamic matching without requiring prior knowledge of the categories number and achieves a 5.04% - 11.16% improvement in AHM. While PA's (Liu et al., 2025) family-level knowledge extraction is limited to datasets with family labels, ConCM captures attribute-level knowledge, enabling finer-grained analysis that achieves a 6.17% performance improvement on CUB200 and generalizes to universal datasets.

## 4.3 ANALYSIS

***ConCM* alleviates knowledge conflict.** Feature confusion between old and novel classes leads to misclassification, which is a tangible manifestation of knowledge conflict. Following Wang et al. (2023), we treat base classes as "positive" and novel classes as "negative", and use the Balanced Error Rate (BER = (FPR + FNR)/2) to quantify conflict. Table 3 reports the metrics for each session, showing that *ConCM* achieves the lowest misclassification rate and the highest novel class accuracy. Compared to Figure 3 (c), Figure 5 demonstrate that our method effectively mitigates false positive classification. Notably, for all increments, the average NAcc of *ConCM* increased by 2.8%, better aligning with practical requirements. Compared to dwelling on old memories, learning new things is essential for adapting to the environment.

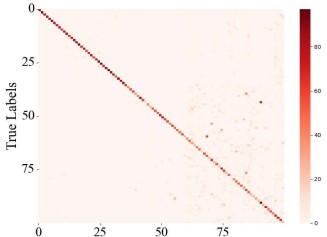

Figure 5: The confusion matrix result on mini-ImageNet.

Table 3: **Analysis of knowledge conflict on mini-ImageNet.** The metric results for eight incremental sessions are reported. **NAcc** denotes the novel-class accuracy, reflecting plasticity of novel classes. **BER** denotes the balanced error rate, reflecting the degree of knowledge conflict. Higher NAcc is better, and lower BER is better.

| Session | 1 | | 2 | | 3 | | 4 | | 5 | | 6 | | 7 | | 8 | |
|---|---|---|---|---|---|---|---|---|---|---|---|---|---|---|---|---|
| NAcc/BER | NAcc | BER | NAcc | BER | NAcc | BER | NAcc | BER | NAcc | BER | NAcc | BER | NAcc | BER | NAcc | BER |
| Baseline(Snell et al., 2017) | 19.80 | 38.20 | 16.90 | 36.53 | 13.53 | 37.09 | 11.10 | 37.22 | 10.80 | 36.06 | 10.00 | 34.99 | 10.03 | 34.44 | 10.57 | 34.78 |
| FACT (Zhou et al., 2022a) | 16.60 | 37.24 | 17.10 | 37.01 | 17.20 | 37.18 | 15.16 | 36.98 | 13.24 | 36.63 | 11.93 | 36.65 | 12.17 | 35.92 | 12.43 | 36.07 |
| CEC (Zhang et al., 2021) | 20.60 | 35.92 | 20.60 | 35.55 | 19.93 | 34.48 | 19.75 | 33.02 | 17.78 | 33.02 | 16.63 | 33.13 | 16.80 | 31.38 | 17.19 | 30.49 |
| TEEN (Wang et al., 2023) | 41.00 | 27.21 | 35.30 | 24.98 | 32.53 | 25.37 | 31.65 | 25.25 | 29.44 | 24.48 | 28.23 | 24.48 | 28.34 | 23.33 | 28.93 | 23.16 |
| OrCo (Ahmed et al., 2024) | 60.20 | 14.63 | 56.90 | 15.12 | 53.40 | 16.48 | 48.45 | 17.87 | 46.40 | 17.55 | 41.70 | 17.85 | 41.43 | 18.13 | 42.55 | 18.56 |
| ConCM | **67.20** | **13.60** | **60.30** | **13.85** | **56.33** | **15.40** | **50.75** | **16.39** | **47.72** | **16.48** | **43.60** | **16.93** | **44.00** | **17.24** | **43.52** | **17.34** |

Table 4: **Ablation studies. AHM** denotes the average harmonic mean. **FA** denotes the accuracy in final session. $\overline{\text{NAcc}}$ denotes average novel-class accuracy. **PD** denotes performance dropping rate. ↑ denotes higher is better. ↓ denotes lower is better. Appendix G presents more results.

| Methods | | | mini-ImageNet | | | | CIFAR100 | | | | CUB200 | | | |
|---|---|---|---|---|---|---|---|---|---|---|---|---|---|---|
| $g(\cdot)$ | MPC | DSM | AHM↑ | FA↑ | $\overline{\text{NAcc}}$↑ | PD↓ | AHM↑ | FA↑ | $\overline{\text{NAcc}}$↑ | PD↓ | AHM↑ | FA↑ | $\overline{\text{NAcc}}$↑ | PD↓ |
| | | | 22.00 | 52.62 | 12.84 | 31.35 | 21.73 | 51.14 | 12.64 | 31.68 | 46.78 | 52.97 | 33.87 | 27.55 |
| ✓ | | | 47.83 | 56.22 | 35.17 | 27.75 | 48.05 | 55.33 | 35.80 | 26.67 | 58.48 | 58.82 | 47.99 | 22.00 |
| ✓ | ✓ | | 52.35 | 57.23 | 40.65 | 26.74 | 53.31 | 56.74 | 42.31 | 25.26 | 59.88 | 59.01 | 50.43 | 21.51 |
| ✓ | | ✓ | 56.79 | 58.29 | 46.81 | 26.68 | 56.96 | 56.99 | 47.95 | 25.83 | 60.19 | 59.59 | 51.43 | 20.93 |
| ✓ | ✓ | ✓ | **59.78** | **59.92** | **51.74** | **24.05** | **59.05** | **58.33** | **51.88** | **24.49** | **62.20** | **62.66** | **53.96** | **17.86** |

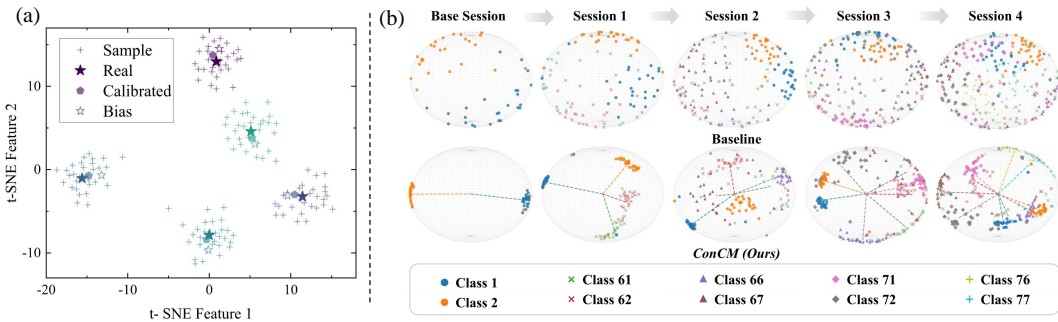

Figure 6: (a) **Prototype calibration visualization.** Different markers indicate the sample feature, the real class center, the calibrated prototype, and the biased prototype. (b) **Embedding space visualization on mini-ImageNet.** Each point represents a sample, colored by class.

**Ablation studies of each module.** Table 4 summarizes the ablation results. Performance Dropping (PD) measures the average accuracy drop from the base to the final session, reflecting forgetting. The baseline uses a frozen backbone, and $g(\cdot)$ denotes a fine-tuned projector. The results show that both the MPC and DSM modules improve performance. For example, on mini-ImageNet, AHM increased by 4.52% and 8.96%, respectively. The integration of both modules entails feature-structure dual consistency, achieving the best performance with an 11.95% improvement.

***ConCM* achieves better feature consistency.** We performed visualization on the features, as shown in Figure 6. Compared to Figure 3 (b), *ConCM* enhances feature representations by semantic associations between different classes, effectively reducing the deviation from the true centers. Based on the calibrated features, we further visualized their embedding structure, as shown in Fig. 6 (b). Compared to the baseline, its embedding is no longer scattered chaotically across the structure, but instead forms a more compact spatial distribution, naturally reducing classification difficulty.

***ConCM* achieves better structure consistency.** To validate the effect of maximum matching in structure consistency, we define the Structure Matching Rate (SMR $= \sum \langle \delta'_i, \delta_i \rangle$) to quantify the deviation between the initial and target structure. The results are shown in Table 5. Random Dynamic Matching (RM) randomly constructs and matches an optimal structure, which disrupts consistency. GM refers to greedy matching (Kuhn, 1955) leading to improvement, but still constrained by the structure itself. Fixed Structure (FS) pre-allocates a structure for all classes, but it leads to a trivial solution. *ConCM* outperforms in all incremental sessions, achieving the highest SMR and HM, while obtaining optimal performance with minimal structure adjustments.

Table 5: **Analysis of structure consistency on mini-ImageNet. HM** denotes harmonic mean accuracy. **SMR** denotes structural matching rate. *ConCM* achieves the best performance by maximal matching relation.

| Session | 1 | | 2 | | 3 | | 4 | | 5 | | 6 | | 7 | | 8 | |
|---|---|---|---|---|---|---|---|---|---|---|---|---|---|---|---|---|
| **HM/SMR** | HM | SMR | HM | SMR | HM | SMR | HM | SMR | HM | SMR | HM | SMR | HM | SMR | HM | SMR |
| RM | 66.03 | 0.01 | 60.90 | 0.01 | 58.62 | 0.02 | 53.04 | 0.02 | 52.77 | 0.01 | 49.45 | 0.01 | 48.89 | 0.01 | 49.88 | 0.02 |
| GM | 66.62 | 0.19 | 61.02 | 0.20 | 57.78 | 0.20 | 54.20 | 0.18 | 52.77 | 0.19 | 49.31 | 0.20 | 49.56 | 0.20 | 50.65 | 0.20 |
| FS(Yang et al., 2023) | 70.10 | 0.85 | 63.44 | 0.65 | 61.95 | 0.69 | 56.01 | 0.73 | 55.83 | 0.74 | 52.74 | 0.78 | 52.66 | 0.81 | 52.76 | 0.80 |
| ConCM | 70.34 | 0.93 | 66.59 | 0.83 | 63.58 | 0.84 | 59.59 | 0.84 | 57.05 | 0.85 | 53.95 | 0.85 | 53.49 | 0.85 | 53.92 | 0.85 |

**Feature-structure dual consistency fosters more robust continual learning.** Features are matched to the optimal structure via loss-driven optimization. To analyze the impact of different losses, we use cross-entropy loss $\mathcal{L}_{ce}$ (*i.e.*, without structure), matching loss $\mathcal{L}_{Match}$ (*i.e.*, structure), general SCL loss $\mathcal{L}_{SCL}$ and contrastive loss with anchor $\mathcal{L}_{Cont}$. We also introduce the intra-class ($Sim_{cls}$) and inter-class ($Sim_{in}$) cosine similarities in the final session to quantify the degree of matching. Table 6 shows that $\mathcal{L}_{Match}$ fosters intra-class cohesion, whereas $\mathcal{L}_{Cont}$ encourages inter-class separation. Structure anchors help the model better capture the structure, resulting in optimal feature-structure matching. The ablation results of the MPC module show that feature inconsistency weakens matching performance.

Table 6: **Ablation on Loss.** w/o denotes without MPC.

| Train Strategy | $Sim_{cls}$ ↓ | $Sim_{in}$ ↑ | $\overline{NAcc}$ ↑ | AHM ↑ |
|---|---|---|---|---|
| $\mathcal{L}_{ce}$ | 0.0366 | 0.3140 | 35.17 | 47.83 |
| $\mathcal{L}_{Match}$ | **0.0111** | 0.3187 | 43.93 | 55.28 |
| $\mathcal{L}_{Match} + \mathcal{L}_{SCL}$ | 0.1233 | **0.4182** | 51.35 | 59.42 |
| $\mathcal{L}_{Match} + \mathcal{L}_{Cont}$ w/o | 0.0406 | 0.3174 | 46.81 | 56.79 |
| $\mathcal{L}_{Match} + \mathcal{L}_{Cont}$ | 0.0285 | 0.3355 | **51.74** | **59.78** |

**ConCM achieves SOTA with less memory, shorter time, and comparable complexity.** For fair comparison, we adopt a uniformly pre-trained backbone. As shown in Table 7, ConCM improves performance with comparable complexity and achieves the lowest time cost, reducing total time by 11% by avoiding repeated backbone propagation. In terms of memory, while OrCo and NC-FSCIL store five samples per class, ConCM reduces overhead by storing only the base-class feature mean and covariance diagonal.

Table 7: **Comparison of Computational Efficiency.**

| Method | Parameter | FLOPS | Time | Memory | AHM |
|---|---|---|---|---|---|
| NC-FSCIL | 13.62M | 4.72G | 18.10min | 24.78M | 52.78 |
| OrCo | 12.49M | 4.72G | 18.34min | 24.78M | 57.30 |
| ConCM | 13.22M | 4.77G | 16.37min | 9.57M | 59.37 |

**ConCM remains competitive when knowledge base coverage is insufficient.** ConCM is a feature-structure joint calibration and matching framework. If the class names are not presented in WordNet's knowledge base, ConCM can solely utilize the DSM module to construct an effective embedding structure. In this scenario, compared to the baseline method NC-FSCIL as shown in Table 8, ConCM still achieved a performance improvement of 3.05% - 9.07% in AHM.

Table 8: **Comparison on Limited Knowledge Base.**

| Method | mini-ImageNet | CIFAR100 | CUB200 |
|---|---|---|---|
| NC-FSCIL | 52.62 | 47.89 | 57.14 |
| ConCM w/o MPC | **56.79** | **56.96** | **60.19** |

## 5 CONCLUSION

FSCIL involves the conflicting objectives of learning novel knowledge from limited data while preventing forgetting. In this study, we further explore two potential causes of knowledge conflict: feature inconsistency and structure inconsistency. To this end, we propose a consistency-driven calibration and matching framework (*ConCM*). Memory-aware prototype calibration ensures conceptual consistency of features by semantic associations, while dynamic structure matching unifies cross-session structure consistency via structure distillation. Experimental results show that the method achieves more robust continual learning by consistent feature-structure optimization.

**Limitations:** The FSCIL methods in this paper typically use most classes (*e.g.*, 60%) as base classes, requiring diversity samples for reliable semantic attributes. Despite this, our experiments show that *ConCM* still alleviates knowledge conflict. Moreover, Future work will explore more realistic incremental tasks, *e.g.*, active learning on streaming data, to overcome dataset constraints.

## ACKNOWLEDGEMENT

This work was supported in part by the National Natural Science Foundation of China (92467107, 62573440), in part by the Scientific Research Innovation Capability Support Project for Young Faculty (ZYGXQNJSKYCXNLZCXM-I27).

## STATEMENTS

**Ethics Statement.** Our study does NOT involve any of the potential issues such as human subject, public health, privacy, fairness, security, etc. All authors of this paper confirm that they adhere to the ICLR Code of Ethics.

**Reproducibility Statement.** For our theoretical result Theorem 1, we offer the proof in Appendix E. All datasets used in this paper are public and have been cited. Please refer to Appendix F for the dataset descriptions and the implementation details of our experiments. Our source code is released at https://github.com/wire-wqz/ConCM

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

# Appendix

## A  THE USE OF LLM

All work in this study was completed solely by the authors without the use of any large language models (LLMs) for content generation.

## B  ADDITIONAL RELATED WORKS

### B.1  CLASS-INCREMENTAL LEARNING

Class-Incremental Learning (CIL) aims to continually learn novel classes from dynamic data distributions while retaining memory of learned classes. Its main challenge is balancing the plasticity for novel class learning and the stability of old class memories. Current mainstream methods can be categorized into five types (Wang et al., 2024): (1) *Regularization-based methods* (Wu et al., 2019; Li & Hoiem, 2018; Yang et al., 2024; Kirkpatrick et al., 2017) constrain key parameters to avoid novel class learning from disrupting old class knowledge; (2) *Replay-based methods* (Tiwari et al., 2022; Rebuffi et al., 2017) alleviate catastrophic forgetting by retaining a subset of old class samples; (3) *Optimization-based methods* (Wang et al., 2021; Liu & Liu, 2022; Foret et al., 2021) explicitly design gradient projections to leverage flat minima, achieving the ideal solution for multi-task learning; (4) *Representation-based methods* (Javed & White, 2019; Beaulieu et al., 2020) enable continual learning by creating and leveraging representational advantages, such as self-supervised learning and pre-training; (5) *Architecture-based methods* (Liang & Li, 2024; Wang et al., 2022; Yan et al., 2021) build task-specific parameters through careful design of the architecture, avoiding interference between tasks. Existing methods fail to account for the knowledge compatibility between tasks, and in few-shot scenarios, they risk overfitting novel class features, hindering effective generalization of new knowledge.

### B.2  FEW-SHOT LEARNING

Few-Shot Learning (FSL) relies on limited labeled data for novel class learning (Wang et al., 2020). Existing methods typically achieve this goal from the perspectives of data generation, metric learning, and meta learning. (1) *Data generation-based methods* enhance few-shot learning by generating samples that resemble the real distribution (Yang et al., 2021; Pan et al., 2024; Zhang et al., 2023); (2) *Metric learning-based methods* (Qi et al., 2018; Snell et al., 2017) focus on learning a unified distance metric and performing feature embedding; (3) *Meta learning-based methods* (Nichol et al., 2018; Finn et al., 2017) train models to quickly adapt to new tasks by simulating the learning process. However, it neglects the issue that class data typically arrives asynchronously, hindering joint training and adaptation to the streaming nature of real-world data.

### B.3  FEW-SHOT CLASS-INCREMENTAL LEARNING

Few-Shot Class-Incremental Learning (FSCIL) combines the two learning paradigms mentioned above, closely resembling human learning. This paradigm requires the model to continually integrate novel classes while preserving learned knowledge and handling the sparse labeling constraints of new classes (Zhang et al., 2025; Ma'sum et al., 2025). Similar to our method, optimization-based methods address the complexities of optimization to overcome the few-shot overfitting of novel classes and catastrophic forgetting of old classes. This includes: (1) *Replay or distillation methods*, which use old class data or model information to preserve knowledge (Tao et al., 2020; Zhang et al., 2021); (2) *Meta learning methods* (Hersche et al., 2022; Zhou et al., 2022b), which draw on experiences from multiple sessions to enhance future performance; (3) *Feature fusion-based methods* (Goswami et al., 2024; Wang et al., 2023; Akyürek et al., 2022) integrate or combine features obtained from different information sources or feature extraction methods to create more comprehensive and effective representations; (4) ***Feature space-based methods*** (Zhou et al., 2022a; Deng & Xiang, 2024; Yang et al., 2023; Ahmed et al., 2024), highly relevant to our method, focus on optimizing the feature space for FSCIL, with the core objective of learning more robust feature representations.

For instance, FACT (Zhou et al., 2022a) creates virtual prototypes to reserve extra space in the embedding space, allowing novel classes to be integrated with reduced interference. NC-FSCIL (Yang et al., 2023) pre-defines a classifier using neural collapse theory and constrains learning through a projection layer, avoiding conflicts between learning novel classes and retaining old class memories. OrCo (Ahmed et al., 2024) constructs a globally orthogonal feature space during the base session and combines feature perturbation with contrastive learning to reduce inter-class interference. However, on one hand, these methods do not account for the inherent bias in few-shot features, and lacks an accurate anchor point (Goswami et al., 2024; Wang et al., 2023). On the other hand, fixed pre-defined structures implicitly constrain novel classes, forcing them to align with a designated space through optimization. This can lead to trivial solutions, making it difficult to achieve the desired outcome. In contrast, we provide calibrated prototype anchors for the structure using semantic attributes with class separability and environmental invariance, and reduce optimization complexity through maximum matching to obtain the optimal feature embedding.

## C  PSEUDO-CODE FOR CONCM

The training procedure for the **ConCM** is described in Algorithm 1

---
**Algorithm 1** Consistency-driven Calibration and Matching framework (ConCM)

---
**Input:** Training set $\mathcal{D}_{train} = \{\mathcal{D}^0, \mathcal{D}^1, \cdots, \mathcal{D}^T\}$
**Output:** Backbone $f(\cdot; \theta_f) \in \mathbb{R}^{d_f}$, projection $g(\cdot; \theta_g) \in \mathbb{R}^{d_g}$
    *Base Session Pretraining*
1: **for** $\forall((x_i, y_i, c_i)) \in \mathcal{D}^0$ **do**
2:       Train and freeze the backbone $f(\cdot; \theta_f)$ by cross-entropy loss $\mathcal{L}_{CE}$
3:       Get semantic knowledge of attribute $\Pi_0 = \{\mathcal{S}_a, \mathcal{F}_a, \mathcal{S}_c^0, \mathcal{R}^0\}$
4:       Train the MPC $h(\cdot; \theta_h) \in \mathbb{R}^{d_f}$ in Eq. (4)
5:       Get an optimal structure $\Delta_0$ in Eq. (9)
6:       Optimize the projection $g(\cdot; \theta_g)$ in Eq. (13) and Eq. (14)
7: **end for**
    *Incremental Session Fast Adaptation*
8: **for** $\forall((x_i, y_i, c_i)) \in \mathcal{D}^t$ **do**
9:       Get semantic knowledge of attribute $\Pi_t = \{\mathcal{S}_a, \mathcal{F}_a, \mathcal{S}_c^t, \mathcal{R}^t\}$
10:      Calibrate prototypes in Eq. (5)
11:      Sample augmented data $Aug(\Omega_t)$ in Eq. (18)
12:      Update the structure $\Delta_t$ in Eq. (8) and Eq. (9)
13:      Optimize the projection $g(\cdot; \theta_g)$ in Eq. (13) and Eq. (14)
14: **end for**
15: **return** Backbone $f(\cdot; \theta_f) \in \mathbb{R}^{d_f}$, projection $g(\cdot; \theta_g) \in \mathbb{R}^{d_g}$

---

## D  DETAILS OF PROTOTYPE AUGMENTATION

According to the FSCIL setting, previous session samples are inaccessible. To address this issue, we propose augmenting the prototype of base classes to replay class distribution. Simultaneously, we augmented the calibration prototype of novel classes. Therefore prototype augmentation further supports structural alignment within embedding space.

Assuming that the features follow a Gaussian distribution $N(p, \Sigma)$ (Goswami et al., 2024; Yang et al., 2021). During the base session, the distribution can be directly estimated owing to sufficient samples, allowing close approximation to the true distribution, denoted as $N(p^{base}, \Sigma^{base})$. Where $p^{base}$ represents the base feature mean, and $\Sigma^{base}$ represents the base covariance diagonal. For novel classes, the feature mean $\hat{p}'_k$ provided by the calibrated prototype generated by the MPC network. However, due to the scarcity of samples for novel classes, it is not possible to accurately estimate their covariance. DC (Yang et al., 2021) found that classes with similar means exhibit similar variances. We further assessed the Pearson correlation between prototype similarity and variance similarity across classes. The results indicate a statistically significant positive correlation between them ($r = 0.94, p = 0.99 \times 10^{-4}$). Therefore, we leverage the distribution information of

the base class to estimate the covariance of novel. Specifically, the covariance estimation process is as follows:

$$\hat{\Sigma}'_k = \beta(\Sigma_k + \sum_{b=1}^{N_0} \omega_{b,k} \Sigma_b^{base}) \tag{16}$$

where, $\beta$ denotes the covariance scaling degree. $\Sigma_k$ denotes novel class covariance diagonal. We incorporate the base-class information via softmax similarity weighting to obtain $\hat{\Sigma}'_k$. The weighting scheme is defined by the following formula:

$$\omega_{b,k} = \frac{\exp(\gamma \cdot \cos(p_b^{base}, \hat{p}'_k))}{\sum_{i=1}^{N_0} \exp(\gamma \cdot \cos(p_i^{base}, \hat{p}'_k))} \tag{17}$$

$\gamma$ controls the sharpness of the weight calculation (with $\gamma = 16$ in all experiments). In the session $t$, the prototype $\Omega_t = \{p_b^{base}\}_{b=1}^{N_0} \cup \{\hat{p}'_k\}_{k=N_0+1}^{N_t}$ is augmented through Gaussian sampling:

$$\begin{cases} Aug(p_b^{base}) \sim N(p_b^{base}, \Sigma_b^{base}), \ 0 < b \leqslant N_0 \\ Aug(\hat{p}'_k) \sim N(\hat{p}'_k, \hat{\Sigma}'_k), \ N_0 < k \leqslant N_t \end{cases} \tag{18}$$

## E  THEORETICAL PROOF

***Definition1 (Geometric Optimality).***  The neural collapse theory reveals an optimal structure formed by the last-layer features and the classifier (Papyan et al., 2020). A structural matrix $\Delta_t = [\delta_1, \delta_2, \cdots, \delta_{N_t}] \in \mathbb{R}^{d_g \times N_t}(d_g > N_t)$ is said to satisfy geometric optimality if the following condition holds.

$$\forall i, j, \ \delta_i^\top \delta_j = \frac{N_t}{N_t - 1}\lambda_{i,j} - \frac{1}{N_t - 1} \tag{19}$$

where $\lambda_{i,j} = 1$ when $i = j$, and 0 otherwise. It implies that prototypes are equidistantly separated.

***Definition2 (Maximum Matching).***  We define the maximum matching relationship as maximizing the similarity between the optimal target structure $\Delta_t$ and the initial incremental structure $\Delta'_t = [\delta'_1, \delta'_2, \cdots, \delta'_{N_t}] \in \mathbb{R}^{d_g \times N_t}$ (the historical structure that incorporates the embedding of novel classes), which can be expressed as:

$$\Delta_t = \arg\max_{\delta_i \in \Delta_t} \sum_{i=1}^{N_t} \langle \delta'_i, \delta_i \rangle \tag{20}$$

The relationship aims to embed new classes with minimal structural changes.

Each incremental session generates augmented samples $Aug(\Omega_t)$ by Gaussian sampling from memory prototypes $\Omega_t$, which include both the base class prototypes and the novel class calibrated prototype. The projected mean of these samples serves as the initial structure $\Delta'_t$. The structure optimization process is formally stated in the following theorem.

***Theorem1.***  For session $t$, the dynamic structure that satisfies geometric optimality $\Delta_t$ and maximum matching can be updated based on the following formulation.

$$W\Lambda V^\top = \Delta'_t \left(I_{N_t} - \frac{1}{N_t}\mathbf{1}_{N_t}\mathbf{1}_{N_t}^\top\right), \quad U_t = WV^\top \tag{21}$$

$$\Delta_t = \sqrt{\frac{N_t}{N_t - 1}} U_t \left(I_{N_t} - \frac{1}{N_t}\mathbf{1}_{N_t}\mathbf{1}_{N_t}^\top\right) \tag{22}$$

where $U_t = [u_1, u_2 \cdots u_{N_t}] \in \mathbb{R}^{d_g \times N_t}$. $I_{N_t} \in \mathbb{R}^{N_t \times N_t}$ is the identity matrix, and $\mathbf{1}_{N_t} \in \mathbb{R}^{N_t \times 1}$ is an all-ones column vector. $W \in \mathbb{R}^{d_g \times N_t}$, $\Lambda \in \mathbb{R}^{N_t \times N_t}$ and $V^\top \in \mathbb{R}^{N_t \times N_t}$ represents the compact SVD result.

***Proof.***  According to the properties of singular value decomposition, $W$ is a column-wise orthogonal matrix, *i.e.*, $W^\top W = I_{N_t}$ and $V$ is an unitary matrix, *i.e.*, $V^\top V = VV^\top = I_{N_t}$. It can be deduced that $U_t$ satisfying:

$$U_t^\top U_t = (WV^\top)^\top WV^\top = VW^\top WV^\top = I_{N_t} \tag{23}$$

Therefore, $U_t$ is also a column-wise orthogonal matrix, satisfying:

$$\begin{cases} \forall i \neq j, u_i^\top u_j = 0; \\ \forall i = j, u_i^\top u_j = 1; \end{cases} \tag{24}$$

$\Delta_t = [\delta_1, \delta_2, \cdots, \delta_{N_t}]$ is the target geometry vector matrix, and $\delta_i$ corresponds to category $i$. According to Eq. (22), $\delta_i$ can be further simplified as:

$$\delta_i = \sqrt{\frac{N_t}{N_t - 1}} \cdot (u_i - \frac{\sum_{n=1}^{N_t} u_n}{N_t}) \tag{25}$$

For any geometric vectors $\delta_i$ and $\delta_j$, we have:

$$\begin{aligned} \forall i, j, \ \delta_i^\top \delta_j &= \frac{N_t}{N_t - 1} \cdot (u_i - \frac{\sum_{n=1}^{N_t} u_n}{N_t})^\top \cdot (u_j - \frac{\sum_{n=1}^{N_t} u_n}{N_t}) \\ &= \frac{N_t}{N_t - 1} \cdot (u_i^\top u_j - \frac{\sum_{n=1}^{N_t} u_n^\top u_j}{N_t} - \frac{\sum_{n=1}^{N_t} u_i^\top u_n}{N_t} + \frac{\sum_{n=1}^{N_t} u_n^\top \sum_{n=1}^{N_t} u_n}{N_t^2}) \\ &= \frac{N_t}{N_t - 1} \cdot (u_i^\top u_j - \frac{1}{N_t}) \\ &= \frac{N_t}{N_t - 1} \lambda_{i,j} - \frac{1}{N_t - 1} \end{aligned} \tag{26}$$

where $\lambda_{i,j} = 1$ when $i = j$, and 0 otherwise. Therefore, we satisfy geometric optimality. The proof conclusion is: when $U_t$ is a column-wise orthogonal matrix (i.e., $U_t^\top U_t = I_{N_t}$), the matrix calculated by Eq. (22) satisfies the geometric optimality defined in Eq. (19).

Furthermore, the maximum matching relationship requires that Eq. (20) be achieved under geometric optimality, which can be expressed as follows:

$$\begin{aligned} \Delta_t &= \arg\max_{\delta_i \in \Delta_t} \sum_{i=1}^{N_t} \langle \delta_i', \delta_i \rangle \\ &\Rightarrow \Delta_t = \arg\max_{\Delta_t} \operatorname{tr}(\Delta_t'^\top \Delta_t) \end{aligned} \tag{27}$$

Substituting the optimal geometry, as constructed in Eq. (22) (with the condition $U_t^\top U_t = I_{N_t}$), into the above equation yields:

$$U_t = \arg\max_{U_t^\top U_t = I_{N_t}} \operatorname{tr} \left( \Delta_t'^\top \sqrt{\frac{N_t}{N_t - 1}} U_t \left( I_{N_t} - \frac{1}{N_t} \mathbf{1}_{N_t} \mathbf{1}_{N_t}^\top \right) \right) \tag{28}$$

We let $k = \sqrt{\frac{N_t}{N_t - 1}}$ and $M = I_{N_t} - \frac{1}{N_t} \mathbf{1}_{N_t} \mathbf{1}_{N_t}^\top$, where $M$ is a symmetric matrix, and it satisfies $M^\top = M$. According to the trace properties, we have:

$$\begin{aligned} U_t &= \arg\max_{U_t^\top U_t = I_{N_t}} k \cdot \operatorname{tr}(\Delta_t'^\top U_t M) \\ &= \arg\max_{U_t^\top U_t = I_{N_t}} k \cdot \operatorname{tr}(M \Delta_t'^\top U_t) \\ &= \arg\max_{U_t^\top U_t = I_{N_t}} k \cdot \operatorname{tr}((\Delta_t' M)^\top U_t) \end{aligned} \tag{29}$$

Substitute the SVD $\hat{W} \hat{\Lambda} \hat{V}^\top = \Delta_t' M$ into the above formula, where $\hat{W} \in \mathbb{R}^{d_g \times d_g}$, $\hat{\Lambda} \in \mathbb{R}^{d_g \times N_t}$, $\hat{V} \in \mathbb{R}^{N_t \times N_t}$. $\hat{U}$ and $\hat{V}$ are unitary matrices, and Eq. (29) can be further simplified to:

$$\begin{aligned} U_t &= \arg\max_{U_t^\top U_t = I_{N_t}} k \cdot \operatorname{tr}(\hat{V} \hat{\Lambda}^\top \hat{W}^\top U_t) \\ &= \arg\max_{U_t^\top U_t = I_{N_t}} k \cdot \operatorname{tr}(\hat{\Lambda}^\top \hat{W}^\top U_t \hat{V}) \end{aligned} \tag{30}$$

Further define $R = \hat{W}^\top U_t \hat{V} \in \mathbb{R}^{d_g \times N_t}$. Since $R^\top R = \hat{V}^\top U_t{}^\top \hat{W} \hat{W}^\top U_t \hat{V} = I_{N_t}$, it follows that $R$ is also a column-wise orthogonal matrix. Substituting into Eq. (30) yields:

$$U_t = \underset{U_t^\top U_t = I_{N_t}}{\arg\max}\ k \cdot tr(\hat{\Lambda}^\top R) \tag{31}$$

Note that the maximum value is only related to $tr(\hat{\Lambda}^\top R)$. It can be simplified to:

$$tr(\hat{\Lambda}^\top R) = \sum_{i=1}^{N_t} r_{ii}\sigma_i \tag{32}$$

$\sigma_i$ is a singular value, $r_{ii}$ is a diagonal element of $R$. Since $R$ is a column-wise orthogonal matrix, its diagonal elements satisfy $|r_{ii}| \leqslant 1$, and it is obvious that the maximum value is obtained when $r_{ii} = 1$. Therefore, we can get:

$$R = \begin{bmatrix} I_{N_t} \\ 0_{(d_g - N_t) \times N_t} \end{bmatrix} \tag{33}$$

where $0_{(d_g - N_t) \times N_t}$ is a zero matrix of size $(d_g - N_t) \times N_t$. Now, substitute $R$ back into the equation for $U_t$ :

$$R = \begin{bmatrix} I_{N_t} \\ 0_{(d_g - N_t) \times N_t} \end{bmatrix} = \hat{W}^\top U_t \hat{V} \Rightarrow U_t = \hat{W} \begin{bmatrix} I_{N_t} \\ 0_{(d_g - N_t) \times N_t} \end{bmatrix} \hat{V}^\top = WV^\top \tag{34}$$

$W$ and $V^\top$ denote the compact SVD results.

# F   MORE DETAILS

## F.1   DATASET DETAILS

We evaluate our ConCM framework on three FSCIL benchmark datasets: mini-ImageNet(Deng et al., 2009) CIFAR100 (Krizhevsky et al., 2009) and CUB200 (Wah et al., 2011). The splitting details (*i.e.*, the class order and the selection of support data in incremental sessions) follow the previous methods (Zhang et al., 2021; Ahmed et al., 2024; Wang et al., 2023).

**mini-ImageNet:** The dataset is a subset of ImageNet, consisting of 100 classes, with each class containing 600 images. We divided it into 60 base classes and 40 incremental classes, structured into 8 incremental sessions with a 5-way, 5-shot FSCIL scenario.

**CIFAR100:** This dataset consists of 60,000 color images (32×32 pixels) across 100 object classes. For the FSCIL task, it is split into 60 base classes and 40 novel classes, organized into 8 incremental sessions under a 5-way, 5-shot setting.

**CUB200:** This dataset is a fine-grained bird classification dataset, consisting of 200 classes and a total of 11,788 images. In the FSCIL task, it includes 100 base classes and 100 incremental classes, arranged into a 10-way, 5-shot task with a total of 10 incremental sessions.

## F.2   ATTRIBUTE SELECTION

For standard datasets, such as mini-ImageNet(Deng et al., 2009) and CIFAR100(Krizhevsky et al., 2009), the class names (e.g., "house finch") can be obtained. We use the "part_meronyms()" provided by WordNet (Ge et al., 2022) to obtain part–whole relationships as candidate attributes, such as "house finch" → "beak", "wing", "feather", etc. For base classes, we directly select the candidate attributes as the actual attributes. Then, the attributes of all base classes are aggregated into an attribute pool for matching with the novel classes. For novel classes, their actual attributes are defined as the intersection between their candidate attributes and the base class attribute pool, enabling prototype calibration through the associated base class attributes.

Certain datasets, such as CUB200, offer explicitly accessible attribute labels for all samples, which we directly utilize. For each class, we select the 28 most frequent attributes (corresponding to 28 distinct attribute categories) as candidates. The attribute pool construction and novel class attribute selection follow the same procedure as described above.

### F.3 MODEL ARCHITECTURE

In the framework of this paper, the learnable network consists of the following components:

**Backbone network.** The symbolic representation is $f(\cdot; \theta_f) \in R^{d_f}$. Prior studies widely adopt ResNet architecture for FSCIL experiments. Table 9 compares the backbone networks used in different studies. Following OrCo (Ahmed et al., 2024), for mini-ImageNet, we use ResNet18. For CIFAR100, we use ResNet12. For CUB200, we use ResNet18 (pre-trained on ImageNet).

**MPC network.** The symbolic representation is $h(\cdot; \theta_h) \in R^{d_f}$. It consists of an encoder, an aggregator and a decoder. The encoder $h_e(\cdot; \theta_h^e) \in R^{d_f/2}$ and decoder $h_d(\cdot; \theta_h^d) \in R^{d_f}$ are MLP layers. Leveraging cross-attention to derive relational weights, the aggregator combines the encoder's outputs as input for the decoder.

**Projector network.** The symbolic representation is $g(\cdot; \theta_g) \in R^{d_g}$. The projector further maps the embedding space to the geometric space to achieve alignment between the features and the geometric structure. The projector consists of a two-layer MLP. $L_2$ normalization is applied both before and after the projection layer to construct a hypersphere-to-hypersphere mapping.

Table 9: **A comparison of backbone networks used in different studies.**

| Methods | mini-ImageNet | CIFAR100 | CUB200 |
|---|---|---|---|
| FACT (Zhou et al., 2022a) | ResNet18 | ResNet12 | ResNet18 |
| CEC (Zhang et al., 2021) | ResNet18 | ResNet20 | ResNet18 |
| C-FSCIL (Hersche et al., 2022) | ResNet12 | ResNet12 | - |
| LIMIT (Zhou et al., 2022b) | ResNet18 | ResNet20 | ResNet18 |
| LCwoF (Kukleva et al., 2021) | ResNet18 | - | - |
| BiDist (Zhao et al., 2023) | ResNet18 | ResNet18 | ResNet18 |
| TEEN (Wang et al., 2023) | ResNet18 | ResNet12 | ResNet18 |
| NC-FSCIL (Yang et al., 2023) | ResNet12 | ResNet12 | ResNet18 |
| OrCo (Ahmed et al., 2024) | ResNet18 | ResNet12 | ResNet18 |
| ConCM(Ours) | ResNet18 | ResNet12 | ResNet18 |

### F.4 EVALUATION DETAILS

**Harmonic Mean accuracy.** Sandard accuracy measures, such as Top-1 accuracy, tend to be skewed in favor of the base classes. Following Ahmed et al. (2024); Wang et al. (2023); Kukleva et al. (2021), we introduce the Harmonic Mean accuracy (HM) to mitigate this bias:

$$HM_t = \frac{2 \times BAcc_t \times NAcc_t}{BAcc_t + NAcc_t} \qquad (35)$$

This metric yields a good value only when both the base class and novel class accuracies remain at high levels. In addition to this, Average Harmonic Mean (AHM) averages the harmonic mean scores across all incremental sessions for a consolidated view.

**Final Accuracy.** Final Accuracy (FA) denotes the Top-1 accuracy of the final session, evaluating the overall accuracy after completing the all sessions:

$$FA = Acc_{-1} \qquad (36)$$

**Performance Dropping rate.** Performance Dropping rate (PD) denotes the Top-1 accuracy dropping between the base session and the finial session, which measures the degree of forgetting:

$$PD = Acc_0 - Acc_{-1} \qquad (37)$$

**Balanced Error Rate.** We treat base classes as the "positive", novel classes as the "negative" and transform the FSCIL problem into a two-class classification task. Balanced Error Rate (BER) is used to quantify the degree of knowledge conflict.

$$FNR = \frac{FN}{TP + FN} \times 100\%, \quad FPR = \frac{FP}{FP + TN} \times 100\% \qquad (38)$$

Table 10: **SOTA comparison on CUB200. AHM** denotes the average harmonic mean. **FA** denotes the Top-1 accuracy in final session.

| Methods | Base Acc | Session-wise Harmonic Mean (%) ↑ | | | | | | | | | | AHM | FA |
|---|---|---|---|---|---|---|---|---|---|---|---|---|---|
| | | 1 | 2 | 3 | 4 | 5 | 6 | 7 | 8 | 9 | 10 | | |
| iCaRL(Rebuffi et al., 2017) | 77.30 | 44.18 | 42.93 | 36.13 | 30.22 | 30.86 | 29.01 | 28.45 | 26.88 | 26.44 | 25.24 | 32.03 | 25.23 |
| IW(Qi et al., 2018) | 67.53 | 40.52 | 39.19 | 36.35 | 37.33 | 37.85 | 37.85 | 35.68 | 35.54 | 37.36 | 38.21 | 37.59 | 46.06 |
| FACT(Zhou et al., 2022a) | 77.23 | 62.16 | 57.86 | 50.47 | 52.21 | 50.58 | 51.55 | 52.01 | 50.29 | 51.74 | 51.58 | 53.05 | 56.39 |
| CEC(Zhang et al., 2021) | 75.64 | 56.17 | 53.16 | 45.67 | 48.45 | 45.41 | 46.29 | 46.31 | 45.16 | 46.28 | 46.01 | 47.89 | 54.21 |
| LIMIT(Zhou et al., 2022b) | 79.63 | 61.11 | 57.19 | 50.70 | 52.40 | 50.52 | 51.40 | 52.78 | 51.36 | 52.50 | 53.56 | 53.35 | 57.81 |
| MetaNC(Ran et al., 2024) | 78.84 | 68.90 | 58.11 | 47.36 | 49.38 | 46.59 | 47.46 | 46.73 | 44.88 | 45.68 | 45.03 | 50.01 | 50.62 |
| SAVC(Song et al., 2023) | 80.00 | 64.29 | 60.89 | 54.56 | 58.27 | 55.03 | 57.92 | 57.10 | 57.33 | 57.36 | 57.72 | 58.05 | 60.82 |
| FACL(Nema & Kurmi, 2025) | 80.88 | 68.33 | 61.21 | 57.25 | 57.38 | 57.26 | 58.54 | 59.76 | 58.11 | 56.99 | 58.22 | 59.30 | 62.37 |
| CLOSER(Oh et al., 2024) | 79.40 | 63.69 | 62.29 | 58.21 | 59.48 | 59.04 | 59.13 | 59.96 | 57.91 | 59.31 | 59.01 | 59.80 | 62.38 |
| BiDist(Zhao et al., 2023) | 75.98 | 63.32 | 56.73 | 50.35 | 53.08 | 49.50 | 49.76 | 50.74 | 48.64 | 50.98 | 50.87 | 52.40 | 56.98 |
| ADBS(Li et al., 2025) | 79.45 | 66.40 | 61.02 | 54.46 | 59.29 | 55.59 | 57.34 | 58.10 | 57.69 | 57.09 | 56.96 | 58.40 | 60.16 |
| TEEN(Wang et al., 2023) | 77.26 | 65.99 | 61.83 | 55.59 | 56.91 | 55.49 | 56.07 | 55.47 | 53.86 | 55.15 | 54.86 | 57.12 | 57.98 |
| Mamba-FSCIL(Li et al., 2024) | 80.90 | **76.94** | 65.33 | 58.07 | 57.31 | 56.47 | 57.99 | 58.33 | 57.26 | 58.36 | 58.95 | 60.50 | 61.67 |
| PA(Liu et al., 2025) | 78.50 | 60.97 | 58.53 | 54.32 | 57.02 | 54.68 | 56.01 | 55.86 | 55.02 | 53.89 | 53.91 | 56.03 | 58.89 |
| NC-FSCIL(Yang et al., 2023) | 80.45 | 71.30 | 57.30 | 55.25 | 57.56 | 54.15 | 56.05 | 56.60 | 55.55 | 54.63 | 55.03 | 57.14 | 59.44 |
| OrCo(Ahmed et al., 2024) | 75.59 | 72.02 | 64.57 | 59.38 | 59.78 | 57.19 | 57.14 | 57.10 | 55.45 | 56.80 | 56.63 | 59.61 | 57.94 |
| **ConCM (Ours)** | 80.52 | 75.24 | **66.04** | **59.78** | **60.10** | **59.48** | **60.24** | **60.78** | **59.35** | **60.57** | **60.39** | **62.20** | **62.66** |
| *Better than second* | | | +0.71 | +0.40 | +0.32 | +0.44 | +1.11 | +0.92 | +1.24 | +1.26 | + 1.38 | **+1.70** | +0.28 |

Table 11: **Comparison on ImageNet-1k. AHM** denotes the average harmonic mean.

| Methods | Base Acc | Session-wise Harmonic Mean (%) ↑ | | | | | | | | AHM |
|---|---|---|---|---|---|---|---|---|---|---|
| | | 1 | 2 | 3 | 4 | 5 | 6 | 7 | 8 | |
| NC-FSCIL(Yang et al., 2023) | 74.70 | 50.79 | 47.02 | 43.01 | 41.45 | 40.53 | 39.57 | 38.43 | 37.31 | 42.26 |
| OrCo(Ahmed et al., 2024) | 74.47 | 53.26 | 41.65 | 39.99 | 41.22 | 40.23 | 39.32 | 38.91 | 37.66 | 41.55 |
| **ConCM (Ours)** | 75.25 | **66.04** | **58.69** | **54.46** | **51.11** | **49.10** | **46.33** | **43.95** | **42.87** | **51.56** |

$$\text{BER} = \frac{\text{FNR} + \text{FPR}}{2} \tag{39}$$

**Structure Matching Rate.** We define the Structure Matching Rate (SMR) to quantify the deviation between the initial and target structure:

$$\text{SMR} = \sum_{i=1}^{N_t} \langle \delta'_i, \delta_i \rangle \tag{40}$$

where, the initial structure $\Delta'_t = \left[\delta'_1, \delta'_2, \cdots, \delta'_{N_t}\right] \in \mathbb{R}^{d_g \times N_t}$ includes the historical structure and incorporates the embedding of novel classes. The target structure $\Delta_t = [\delta_1, \delta_2, \cdots, \delta_{N_t}] \in \mathbb{R}^{d_g \times N_t}$ is the optimal structure that satisfies Eq. (6).

### F.5 TRAINING DETAILS

Following previous methods (Kukleva et al., 2021; Cheraghian et al., 2021; Ahmed et al., 2024; Yang et al., 2023; Zhao et al., 2023), we maintain some exemplars from previously seen classes: 5 exemplars are saved for novel classes, while only prototypes are maintained for base classes. Standard image augmentation strategies were applied, including random cropping, random horizontal flipping, random grayscale processing, and random color jittering. The optimizer used is SGD, with cosine scheduling and a warmup period for a few epochs during training. All experiments were conducted on a single RTX 4090 GPU.

We implemented meta-training for the MPC network by constructing a series of 5-shot prototype completion tasks. Structural anchors are only applied to novel classes, as base classes typically have sufficient representation. The prototype augmentation strategy resamples augmented samples in each epoch. 100 and 50 samples were generated for base and novel classes, respectively. Each image was augmented 10 times using image augmentation to balance the dataset. Hyperparameters remain consistent: $\tau = 0.07$, $\gamma = 16$, $\beta = 0.6$, and the MPC network's maximum learning rate is 1. For mini-ImageNet, projector's maximum learning rate is $1e^{-2}$, with $\alpha = 0.6$. For CIFAR100,

Table 12: **Ablation study of modules on mini-ImageNet. AHM** denotes average harmonic mean. **FA** denotes the accuracy in final session. The baseline uses frozen backbone. $g(\cdot)$ denotes learnable projector. MPC denotes the Memory-aware Prototype Calibration module. DSM denotes the Dynamic Structure Matching module.

| $g(\cdot)$ | MPC | DSM | Session-wise Harmonic Mean (%) ↑ | | | | | | | | AHM | FA |
|---|---|---|---|---|---|---|---|---|---|---|---|---|
| | | | 1 | 2 | 3 | 4 | 5 | 6 | 7 | 8 | | |
| | | | 31.91 | 28.00 | 23.21 | 19.53 | 19.06 | 17.80 | 17.84 | 18.70 | 22.00 | 52.62 |
| ✓ | | | 62.12 | 53.75 | 51.13 | 46.76 | 44.22 | 41.59 | 41.33 | 41.80 | 47.83 | 56.22 |
| ✓ | ✓ | | 66.11 | 57.05 | 54.77 | 53.35 | 50.06 | 46.49 | 44.86 | 46.14 | 52.35 | 57.23 |
| ✓ | | ✓ | 68.77 | 63.06 | 59.09 | 56.26 | 53.76 | 50.45 | 51.32 | 51.63 | 56.79 | 58.29 |
| ✓ | ✓ | ✓ | **70.34** | **66.59** | **63.38** | **59.59** | **57.05** | **53.95** | **53.49** | **53.92** | **59.78** | **59.92** |

Table 13: **Ablation study of modules on CIFAR100. AHM** denotes average harmonic mean. **FA** denotes the accuracy in final session. The baseline uses frozen backbone. $g(\cdot)$ denotes learnable projector. MPC denotes the Memory-aware Prototype Calibration module. DSM denotes the Dynamic Structure Matching module.

| $g(\cdot)$ | MPC | DSM | Session-wise Harmonic Mean (%) ↑ | | | | | | | | AHM | FA |
|---|---|---|---|---|---|---|---|---|---|---|---|---|
| | | | 1 | 2 | 3 | 4 | 5 | 6 | 7 | 8 | | |
| | | | 30.55 | 24.75 | 23.47 | 20.27 | 20.01 | 18.81 | 17.54 | 18.44 | 21.73 | 51.14 |
| ✓ | | | 65.87 | 56.43 | 48.62 | 42.83 | 43.89 | 43.64 | 42.92 | 40.18 | 48.05 | 55.33 |
| ✓ | ✓ | | 69.64 | 62.56 | 54.88 | 51.25 | 48.31 | 48.97 | 45.75 | 45.12 | 53.31 | 56.74 |
| ✓ | | ✓ | 70.67 | 64.51 | 58.61 | 55.44 | 54.03 | 52.82 | 50.63 | 48.95 | 56.96 | 56.99 |
| ✓ | ✓ | ✓ | **72.27** | **67.33** | **60.09** | **57.08** | **54.93** | **55.21** | **52.95** | **52.51** | **59.05** | **58.33** |

Table 14: **Ablation study of modules on CUB200. AHM** denotes the average harmonic mean. **FA** denotes the accuracy in final session. The baseline uses a frozen backbone. $g(\cdot)$ denotes a learnable projector. MPC denotes the Memory-aware Prototype Calibration. DSM denotes the Dynamic Structure Matching.

| $g(\cdot)$ | MPC | DSM | Session-wise Harmonic Mean (%) ↑ | | | | | | | | | | AHM | FA |
|---|---|---|---|---|---|---|---|---|---|---|---|---|---|---|
| | | | 1 | 2 | 3 | 4 | 5 | 6 | 7 | 8 | 9 | 10 | | |
| | | | 60.14 | 53.00 | 45.56 | 45.64 | 44.08 | 45.75 | 44.64 | 42.13 | 43.36 | 43.46 | 46.78 | 52.97 |
| ✓ | | | 71.89 | 63.62 | 57.33 | 59.20 | 56.23 | 56.18 | 55.76 | 54.14 | 55.60 | 54.89 | 58.48 | 58.82 |
| ✓ | ✓ | | 73.78 | 65.98 | 59.75 | 59.67 | 57.28 | 57.41 | 58.00 | 55.91 | 55.61 | 55.43 | 59.88 | 59.01 |
| ✓ | | ✓ | 74.06 | 65.56 | 59.03 | **60.34** | 57.49 | 58.25 | 57.35 | 56.37 | 57.10 | 56.33 | 60.19 | 59.59 |
| ✓ | ✓ | ✓ | **75.24** | **66.04** | **59.78** | 60.10 | **59.48** | **60.24** | **60.78** | **59.35** | **60.57** | **60.39** | **62.20** | **62.66** |

projector's maximum learning rate is $2e^{-2}$, with $\alpha = 0.6$. For the Cub200, projector's maximum learning rate is $5e^{-3}$, with $\alpha = 0.75$.

## G MORE RESULTS

**Experimental comparison with State-of-the-Art methods.** Our experiment results on mini-ImageNet, CUB200,and CIFAR100 are shown in Table 1, Table 2 , and Table 10 (Appendix G), respectively. Experimental results show that ConCM consistently maintains the highest performance across all incremental sessions, effectively mitigating the conflict between learning and forgetting. In addition, on ImageNet-1k benchmark, ConCM achieved a 9.30% improvement on AHM compared to the baseline methods (e.g. OrCo, NC-FSCIL) as show in Table 11. The number of categories in the ImageNet-1k benchmark has significantly increased (1,000 classes), which includes 600 base classes and is set up in 8 incremental sessions with a 50-way 5-shot configuration.

**Session-wise ablation study of each module.** In this appendix, we further report the session-wise harmonic mean accuracy for in Table 12, Table 13 and Table 14, respectively. The proposed method achieves consistent performance improvements in each session across all datasets. These results further validate the effectiveness of each module.

**Ablation within the MPC module.** The MPC module includes prototype calibration and Gaussian-based prototype augmentation. To further demonstrate the effectiveness of prototype augmenta-

Table 15: **Ablation on Prototype Augmentation and Calibration within the MPC module.**
Case1: Without prototype calibration. Case2: With prototype calibration. Standard represents standard Gaussian sampling. Ours represents Gaussian sampling-based prototype augmentation.

| Case | Method | Session-wise Harmonic Mean (%) ↑ | | | | | | | | AHM |
|---|---|---|---|---|---|---|---|---|---|---|
| | | 1 | 2 | 3 | 4 | 5 | 6 | 7 | 8 | |
| 1 | Standard | 68.77 | 63.06 | 59.09 | 56.26 | 53.76 | 50.45 | 51.32 | 51.63 | 56.79 |
| | Ours | 70.12 | 63.68 | 60.22 | 56.87 | 55.10 | 52.23 | 52.57 | 52.78 | 57.95 |
| 2 | Standard | 69.83 | 64.40 | 61.42 | 57.36 | 54.24 | 51.17 | 51.45 | 51.80 | 57.71 |
| | Ours | 70.34 | 66.59 | 63.38 | 59.59 | 57.05 | 53.95 | 53.49 | 53.92 | 59.78 |

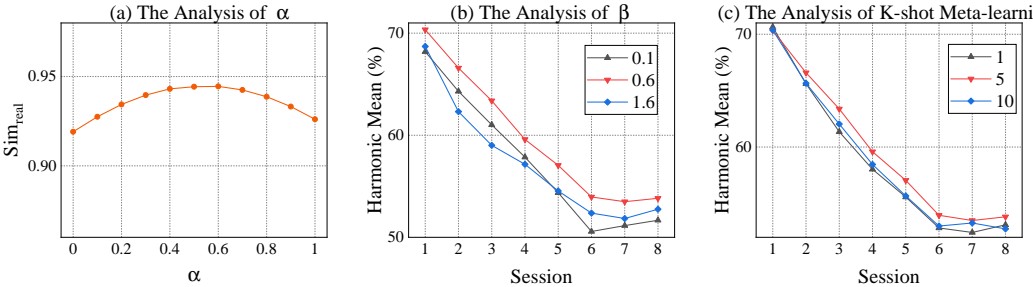

Figure 7: **The hyperparameter analysis on mini-ImageNet.** (a) Prototype calibration degree $\alpha$. (b) Covariance scaling degree $\beta$. (c) Meta-learning under the K-shot scenario tasks.

tion, we compare two cases on mini-ImageNet. Case1: Without prototype calibration, we compare standard Gaussian sampling with Gaussian sampling-based prototype augmentation (ours). Case2: With prototype calibration, (comparison as in Case1). Table 15 presents the experimental results. The within-case comparisons show that prototype enhancement brings performance improvements of 1.16% and 2.07% under the two cases, respectively. The between-case comparisons indicate that prototype calibration leads to performance gains of 0.92% and 1.83%, respectively.

**Hyperparameter analysis.** We conduct an analysis of critical hyperparameters in the ConCM on mini-ImageNet, which include prototype calibration degree $\alpha$, covariance scaling degree $\beta$, and meta-learning under the K-shot scenario tasks.

- **The impact of calibration degree** $\alpha$ is evaluated using the average cosine similarity $Sim_{real}$ between all calibrated prototypes and the real prototype (Figure 7 (a)). Prototype calibration combines the MPC output prototype with the few-shot prototype, both of which exhibit biases: the former due to base-novel class discrepancy, the latter due to limited samples. Selecting appropriate weighting improves the representativeness of the final prototypes.

- **The covariance scaling degree** $\beta$ controls the dispersion of the novel-class distribution, as shown in Figure 7 (b). Appropriate scaling helps restore the real feature distribution.

- **The $K$-shot task construction** employs an episodic meta-training strategy. We analyze the impact on the mini-ImageNet. Generally, a smaller $K$ indicates that prototypes during training exhibit stronger bias. Selecting the appropriate level of difficulty for training can help to improve performance.

Across all benchmark datasets, our hyperparameters remain consistent with $\alpha = 0.6$, $\beta = 0.6$, and $K = 5$, except for CUB200 where $\alpha = 0.75$. This is because CUB200 is a fine-grained dataset with smaller inter-class differences.

**Analysis of replay strategies.** Five exemplars from seen incremental class have been replayed. However, even without replaying samples and instead storing incremental class prototypes and covariance diagonal, our method still maintains high performance. To validate this, we tested with 0, 1, and 5 replay settings, and the results are shown in Table 16. This demonstrates that even without explicit sample replay, using prototypes for augmentation can still effectively mitigate catastrophic forgetting and maintain high performance.

Table 16: **Performance comparison under different replay exemplar numbers.** Without replaying raw samples (#Replay=0) can still maintain high performance effectively.

| #Replay | Session-wise Harmonic Mean (%) ↑ | | | | | | | | AHM |
|---|---|---|---|---|---|---|---|---|---|
| | 1 | 2 | 3 | 4 | 5 | 6 | 7 | 8 | |
| 0 | 70.34 | 65.83 | 61.35 | 58.28 | 55.94 | 52.40 | 52.19 | 52.77 | 58.64 |
| 1 | 70.34 | 65.33 | 61.59 | 58.54 | 56.43 | 52.87 | 52.91 | 53.15 | 58.90 |
| 5 | 70.34 | 66.59 | 63.38 | 59.59 | 57.05 | 53.95 | 53.49 | 53.82 | 59.78 |

Table 17: **Performance comparison under different K-shot and Cross-Domain.** K-shot indicates that the number of samples in each nove class. "Cross-Domain" represents the constructed cross-domain FSCIL task. The base classes are from the base classes of mini-ImageNet, and the novel classes are from the novel classes of CIFAR100, forming an 8-session 5-way 5-shot task.

| Setting | Method | Session-wise Harmonic Mean (%) ↑ | | | | | | | | AHM |
|---|---|---|---|---|---|---|---|---|---|---|
| | | 1 | 2 | 3 | 4 | 5 | 6 | 7 | 8 | |
| 5-way,1-shot | OrCo | 37.78 | 29.66 | 30.83 | 30.50 | 30.06 | 28.96 | 29.08 | 29.08 | 30.74 |
| | ConCM | **54.91** | **45.98** | **40.12** | **37.91** | **32.96** | **32.44** | **31.75** | **31.43** | **38.43** |
| 5-way,3-shot | OrCo | 61.95 | 54.87 | 53.04 | 49.68 | 47.25 | 45.94 | 45.73 | 47.09 | 50.69 |
| | ConCM | **66.52** | **61.31** | **55.74** | **52.25** | **49.26** | **45.96** | **46.56** | **47.56** | **52.97** |
| Cross-Domain | OrCo | **70.52** | 57.01 | 48.99 | 44.98 | 44.86 | 45.23 | 42.31 | 41.36 | 49.41 |
| | ConCM w/o MPC | 68.37 | 56.56 | 49.22 | 46.10 | 44.90 | 45.69 | 43.33 | 43.23 | 49.68 |
| | ConCM | 69.56 | **57.10** | **49.55** | **47.02** | **47.85** | **48.16** | **45.62** | **45.36** | **51.28** |

Table 18: **Performance comparison under long sequence FSCIL on mini-ImageNet.** 2-way, 5-shot setting constructed a 20-sessions task. The table reports the harmonic accuracy of the first 4 sessions and the last 4 sessions.

| Setting | Method | Session-wise Harmonic Mean (%) ↑ | | | | | | | | AHM |
|---|---|---|---|---|---|---|---|---|---|---|
| | | 1 | 2 | 3 | 4 | 17 | 18 | 19 | 20 | |
| 2-way,5-shot | OrCo | 78.91 | 63.29 | 65.30 | 63.51 | **51.05** | 51.38 | 51.15 | 50.97 | 57.01 |
| | ConCM | **78.98** | **67.44** | **69.71** | **66.71** | 50.92 | **51.98** | **52.59** | **52.11** | **58.74** |

**Comparative experiments under various FSCIL settings.** We evaluated multiple N-way K-shot configurations on mini-ImageNet against the suboptimal method OrCo (Ahmed et al., 2024). The results for different K-shot are presented in Table 17. ConCM more effectively mitigates the knowledge conflict and demonstrates stronger robustness in more challenging few-shot settings, achieving a 7.69% AHM improvement under 5-way 1-shot. We further compared methods under long-sequence (20 sessions) FSCIL with varying N-way settings. The performance of the first 4 sessions, the last 4 sessions are reported in Table 18. The proposed method surpasses the OrCo by 4.41% (the highest) in harmonic accuracy of incremental sessions. This verifies the scalability of our approach.

**ConCM's Generalization Across Domains.** ConCM evaluates correlations between novel and base classes through semantic attributes and learns an optimal geometric embedding, enabling certain cross-domain generalization. In our cross-domain experiment with base classes from mini-ImageNet and novel classes from CIFAR100, ConCM outperforms the suboptimal approach, OrCo (in Table 17). We hypothesize that despite the domain shift, latent semantic relationships persist between classes, and ConCM effectively leverages these to reduce negative bias. Moreover, the geometrically aligned embedding derived from maximum matching exhibits strong domain invariance, further enhancing generalization. Additionally, the MPC module achieved a 1.60% improvement in the AHM metric, indicating that it continues to perform effectively even under conditions of weaker semantic relevance.

