# OpenReview forum: "Consistency-Driven Calibration and Matching for Few-Shot Class Incremental Learning"
_ICLR.cc/2026/Conference — ICLR 2026 Poster_

### Official Review · Reviewer_Tr9T · 2025-10-28

**Soundness:** 3
**Presentation:** 2
**Contribution:** 2
**Rating:** 4
**Confidence:** 5

**Summary:**

This paper proposes a consistency-driven calibration-and-matching approach that calibrates biased new-class prototypes and stabilizes the feature-space structure, thereby reducing confusion between old and new classes. The approach demonstrates strong performance across several benchmarks.

**Strengths:**

1. The motivations are clear. Biased new-class distributions caused by limited training samples, and adjustments to the feature space to accommodate new classes, are key challenges in FSCIL, as recognized by the community.
2. Leveraging common attributes to augment the features of new classes is valid and reasonable; WordNet provides rich semantic information beyond visual cues.
3. The proposed approach achieves strong performance on several datasets, including mini-ImageNet, CIFAR-100, and CUB-200.

**Weaknesses:**

1. The idea of “Attribute Separation” is similar to PA [1], which decouples common attributes within a family and transfers them to new species. It would be helpful to discuss the commonalities and differences between your approach and PA [1].
[1] Prototype antithesis for biological few-shot class-incremental learning.
2. Neural collapse theory has been introduced into the FSCIL task by NC-FSCIL. The dynamic structure matching in this paper builds on that work; for example, it replaces the hard enforcement of a computed prototype distribution with a softer approach that reduces the distance between the original and optimal distributions. While this is indeed more reasonable and effective, the core idea and methodology are not fundamentally different from NC-FSCIL; in my view, this is an incremental improvement.
3. When performing prototype augmentation, this paper makes a strong assumption, namely, that feature distributions are Gaussian. However, many real-world class distributions do not align well with this assumption. Furthermore, based on this assumption, the authors infer the covariance of new classes by assuming that classes with similar means also have similar covariances. Is there theoretical justification for this assumption?
4. The MPC module appears highly dependent on the base-class distribution and on WordNet. If the base classes differ substantially from the new classes, or if the names of the new classes are not present in WordNet’s knowledge base, the model’s generalization ability may be greatly affected.
5. There are a few typos. For example, “mini-IamgeNet” at line 432 and “SOAT” at line 360 (these should be “mini-ImageNet” and “SOTA”).

**Questions:**

Please refer to the 'weaknesses'. I would consider to raise my rating if the authors could address my concerns.

---

> ### Author Response · Authors · 2025-11-21
> **The responses to Reviewer Tr9T 1/2**
>
> > **W1: It would be helpful to discuss the commonalities and differences between your approach and PA [1].**
>
> **A1:** Thanks for your constructive comments. We hope the following analysis of the commonalities and differences between the MPC module in ConCM and PA [1] will helpful.
>
> **Commonality:**
>
> Both methods leverage decoupled shared knowledge to refine the modeling of novel class prototypes.
>
> **Difference:**
>
> - **Different purposes.**
>
>    - PA transfers knowledge to novel-class prototypes and performs classification based on the distance to the prototypes.
>
>    - After MPC complements the novel class prototype, it embeds the novel class features into the constructed dynamic structure, with the objective of achieving feature-structure alignment.
>
> - **Different hierarchical levels of shared knowledge.**
>
>   - PA extracts family-level shared knowledge, where each category belongs to only one family. Since this design limits its application to biological datasets with family labels.
>
>   - MPC operates by extracting attribute-level shared knowledge, where a single class can be associated with a variety of attributes. This enables the capture of finer-grained or cross-family characteristics, allowing for generalization to common datasets.
>
> - **Different transfer mechanism.**
>
>   - PA directly fuses the shared features with the novel class prototypes through a weighted summation.
>
>    - MPC aggregates attributes in the latent space and evaluates the association between categories and attributes based on cross-attention.
>
> - **ConCM offers superior performance.**
>
>   - In the CUB200 dataset, we conducted a detailed comparative analysis between ConCM and PA. **The MPC module solely achieved a 3.85% performance** improvement in AHM over PA. Overall, **ConCM delivered a 6.17% performance** boost.
>   |Methods|Base|1|2|3|4|5|6|7|8|9|10|AHM|
>   | :--- | :--- | :--- | :--- | :--- | :--- | :--- | :--- | :--- | :--- | :--- | :--- | :--- |
>   |PA[1]|78.50|60.97|58.53|54.32|57.02|54.68|56.01|55.86|55.02|53.89|53.91|56.03|
>   |ConCM(only MPC)|80.52|73.78|65.98|59.75|59.67|57.28|57.41|58.00|55.91|55.61|55.43|59.88|
>   |ConCM|80.52|75.24|66.04|59.78|60.10|59.48|60.24|60.78|59.35|60.57|60.39|62.20|
>
> [1] Prototype antithesis for biological few-shot class-incremental learning.
>
> ---
> > **W2: The core idea and methodology are not fundamentally different from NC-FSCIL.**
>
> **A2:** Thanks for your feedback. Please allow us to elaborate on the fundamental differences between ConCM and NC-FSCIL.
>
> - **ConCM focuses on feature-structure dual consistency**, a property not considered by other methods. **NC-FSCIL is a static structure optimization** method that pre-assigns a fixed ETF structure as the optimization target, without considering the inherent bias of the embedded features. The core of ConCM lies in jointly optimizing features and structures to dynamically adapt to each incremental task. Our objective is to resolve feature-structure inconsistencies, which fundamentally differs from NC-FSCIL.
>
> - **ConCM has greater adaptability and enhanced performance.** **NC-FSCIL is a closed-set classification** algorithm, where its predefined structure assumes the number of emerging categories is known beforehand. **DSM in ConCM** achieves structural dynamic matching based on the derived maximum matching relationship, **without requiring prior** knowledge of the categories number. More importantly, when considering DSM alone, we achieve **better AHM performance (3.05%~9.07%)**.
> |Methods|mini-ImageNet|CIFAR100|CUB200|
> | :--- | :---: | :---: | :---: |
> |NC-FSCIL|52.62|47.89|57.14|
> |ConCM w/o MPC|56.79|56.96|60.19|

---

> ### Author Response · Authors · 2025-11-21
> **The responses to Reviewer Tr9T 2/2**
>
> > **W3: The hypotheses that features are Gaussian-distributed and classes sharing similar means have comparable covariances, including their theoretical rationale.**
>
> **A3:** Thanks for your insightful question. The Gaussian distribution possesses a straightforward mathematical representation and has been adopted as a common choice in mainstream benchmarking methods [1][2][3], hence we follow this setting. It is widely recognized that the Gaussian assumption is reasonable in the context of FSCIL.
>
> Furthermore, DC[4] observe that **similar classes usually have similar mean and variance of the feature representations**. The mean and variance of the Gaussian distribution can be transferred across similar classes. They demonstrated the effectiveness of covariance transfer from the experimental and the **theory derivation about Generalization error bound**.
>
> [1] Learnable Distribution Calibration for Few-Shot Class-Incremental Learning, TPAMI.
>
> [2] Neural Collapse Inspired Feature-Classifier Alignment for Few-Shot Class Incremental Learning, ICLR.
>
> [3] Calibrating Higher-Order Statistics for Few-Shot Class-Incremental Learning with Pre-trained Vision Transformers, CVPR.
>
> [4] Bridging the Gap Between Few-Shot and Many-Shot Learning via Distribution Calibration, TPAMI.
>
> ---
> > **W4：The MPC module appears highly dependent on the base-class distribution and on WordNet.**
>
> **A4:** Thanks your insightful comment. We are grateful for the opportunity to elucidate the generalization capability of ConCM, particularly in terms of semantic similarity and the coverage of semantic knowledge.
>
> - **ConCM maintains performance in weakly semantic scenarios.** In cross-domain configurations (Table 15, Appendix G), we have considered scenarios where the base class and novel class **exhibit significant semantic differences**. In the table below, we further supplemented the ablation analysis under this setting, where **the MPC module achieved a 1.60% improvement** in AHM, indicating its continued effectiveness even under weak semantic relevance.
> |Methods|1|2|3|4|5|6|7|8|AHM|
> | :--- | :--- | :--- | :--- | :--- | :--- | :--- | :--- | :--- | :--- |
> |ConCM w/o MPC|68.37|56.56|49.22|46.10|44.90|45.69|43.33|43.23|49.68|
> |ConCM|69.56|57.10|49.55|47.02|47.85|48.16|45.62|45.36|51.28|
>
> - **ConCM remains competitive when knowledge base coverage is insufficient.** ConCM is a feature-structure joint calibration and matching framework. If the names of the new class are not presented in WordNet’s knowledge base, ConCM can solely utilize the DSM module to construct an effective embedding structure. In this scenario, compared to the baseline method NC-FSCIL, ConCM still **achieved a performance improvement of 3.05% ~ 9.07%** in AHM.
> |Methods|mini-ImageNet|CIFAR100|CUB200|
> | :--- | :---: | :---: | :---: |
> |NC-FSCIL|52.62|47.89|57.14|
> |ConCM w/o MPC|56.79|56.96|60.19|
>
> ---
> > **W5: There are a few typos.**
>
> **A5:** Thanks for your careful review. It has been corrected, and we've conducted a full proofread to ensure no similar issues remain.

---

> > ### Comment · Reviewer_Tr9T · 2025-11-26
> >
> > Thanks to the authors' efforts, their responses have addressed most of my concerns, specifically regarding the differences between PA and ConCM, the improvements over NC-FSCIL, and the generalization ability to classes not present in WordNet.
> > However, I still have a question concerning the transferability of the Gaussian distribution assumptions. The authors stated in their responses that 'DC[4] observe that similar classes usually have similar mean and variance of the feature representations.' However, this observation is insufficient to directly conclude that two classes with similar prototypes (means) must also possess similar variance. This is not a reversible logical implication. Do the authors have any more conclusive theoretical grounding or experimental evidence to substantiate this specific point?
> > Besides, I suggest the authors incorporate the detailed discussion regarding the fundamental differences between PA and NC-FSCIL, as well as the results of the additional expanded experiments, into the revised paper. This inclusion is crucial to ensure the completeness and rigor of the manuscript.

---

> ### Author Response · Authors · 2025-11-26
> **Further responses to Reviewer Tr9T**
>
> Thanks for your valuable time and response. To address your inquiry regarding the transferability of Gaussian distributions, we designed validation experiments concerning category means and variances based on the settings in DC [1]. The experiment evaluated the correlation between prototype similarity and variance similarity across class, and calculated their Pearson correlation coefficient. **The results revealed a significant positive correlation between them, i.e. $r = 0.94,p = 0.99 \times {10^{ - 4}}$, providing empirical support for the transferability of Gaussian distributions.** Specifically,
>
> 1. We conducted an analysis on the mini-ImageNet dataset, calculating the mean cosine similarity and variance cosine similarity for all category pairs. Subsequently, we provided quantitative metrics based on Pearson correlation analysis.
>
> 2. The specific metrics of the Pearson correlation coefficient are shown in the table below. The results indicate a stable positive correlation between the two similarity measures (mean_sim and var_sim). Additionally, we have listed the similarity trend for some categories, further verifying that for two categories, the closer their means are, the closer their covariances tend to be.
>
> 3. The aforementioned phenomenon occurs because: the mean of features represents the general appearance of a category, while variance represents the deviation range of attribute values. Categories with similar means are highly likely to share similar general appearances, leading to similar ranges of attribute variation [1][2].
>
> - **Pearson correlation coefficient:**
> |Metric| Value|
> | :---| :--- |
> | $r$ | $0.94$|
> | $p$| $0.99 \times10^{ - 4}$ |
>
> - **Similarities relative to arctic fox:**
> |arctic fox|white wolf|malamute| lion | ladybug | ant | house finch | robin | beer bottle |
> | :---: | :---: | :---: | :---: | :---: | :---: | :---: | :---: | :---: |
> |mean_sim|92.70%|81.36%|72.45%|64.49%|64.17%|53.33%|52.30%|49.69%|
> |var_sim|94.49%|86.17%|81.40%|75.14%|76.30%|68.71%|67.46%|66.27%|
>
> - **Similarities relative to robin:**
> | robin | house finch | ladybug |ant| lion|white wolf|malamute |arctic fox| beer bottle |
> | :---: | :---: | :---: | :---: | :---: | :---: | :---: | :---: | :---: |
> |mean_sim|72.41%|66.11%|62.74%|55.96%|54.32%|53.64%|52.30%|46.53%|
> |var_sim| 82.75%| 77.03%| 75.74%| 68.77%| 67.99%| 66.99%| 67.46%| 63.69%|
>
> - **Similarities relative to ant:**
> | ant| ladybug | lion | malamute | house finch |white wolf| arctic fox| robin | beer bottle |
> | :---: | :---: | :---: | :---: | :---: | :---: | :---: | :---: | :---: |
> |mean_sim|68.98%|68.94%|68.41%|66.26%|65.46%|64.17%|62.74%|51.44%|
> |var_sim|79.53%|79.27% |77.71% |78.68% |76.77% |76.30%| 75.74% |68.43%|
>
> We apologize for any confusion caused by the imprecise wording in the manuscript appendix. To address this, we will include a more detailed explanation of the covariance transfer in Gaussian distributions in the final version.
>
> Additionally, we have supplemented the discussion on the fundamental differences between PA and NC-FSCIL in the Related Work and Section 4.2. Extended experiments regarding the generalization ability to classes not present in WordNet have been added to Section 4.3. In the final version, we will further strengthen the aforementioned discussions. We truly appreciate your insights and the time you've dedicated to improving our work. If any issues remain, we would be grateful for your feedback.
>
> [1] Bridging the Gap Between Few-Shot and Many-Shot Learning via Distribution Calibration
>
> [2] Free Lunch for Few-Shot Learning: Distribution Calibration

---

> ### Comment · Reviewer_Tr9T · 2025-11-26
>
> Thanks for the further clarification, and the additional experiment results make the hypotheses more convincing. I'm generally satisfied with the responses, and I'll update my rating to 6, good luck!

---

> > ### Author Response · Authors · 2025-11-26
> > **Thanks for increasing to 6!**
> >
> > Thank you so much for your effort in the review phase and for increasing the score to 6! We will incorporate the above discussion and additional experimental results into the final version. Your valuable feedback greatly improved our paper! Thank you again for your blessings.

---

### Official Review · Reviewer_CEGG · 2025-10-31

**Soundness:** 3
**Presentation:** 3
**Contribution:** 3
**Rating:** 6
**Confidence:** 4

**Summary:**

The paper investigates how to maintain representation consistency in Few-Shot Class-Incremental Learning (FSCIL). The authors propose a framework that calibrates new class prototypes inspired by human associative memory. The calibration module separates and completes semantic attributes to refine prototype representations for new classes, ensuring better alignment. Furthermore, a geometric optimization strategy is introduced to preserve structural consistency during incremental updates. Experimental results on multiple FSCIL benchmarks demonstrate consistent improvements over existing methods.

**Strengths:**

1. The paper is well written, and the motivation is clear.
2. The proposed consistency-driven dynamic structure matching method is theoretically grounded and achieves excellent performance.
3. The ablation experiments are comprehensive and convincing.

**Weaknesses:**

1. The paper missed several important and highly related references and comparative results.

    [1]  Mamba-FSCIL: Dynamic Adaptation with Selective State Space Model for Few-Shot Class Incremental Learning.

    [2] Learning With Fantasy: Semantic-Aware Virtual Contrastive Constraint for Few-Shot Class-Incremental Learning.

    [3] Learning optimal inter-class margin adaptively for few-shot class-incremental learning via neural collapse-based meta-learning.

    [4] Towards Better Representation Learning for Few-Shot Class-Incremental Learning

2. The pipeline of novel class prototypes calibration is the same as paper [1] except for the design of MPC network (both adopt the encode–aggregate–decode architecture). The novelty of this component should be further clarified.

    [1] Prototype completion for few-shot learning.

3. More recent research on CIL mainly focuses on pre-trained ViT or CLIP models. Can the proposed method be transferred or adapted to pre-trained ViT models or CLIP models?

   [1] Pre-trained Vision and Language Transformers Are Few-Shot Incremental Learners.

**Questions:**

1. It is unclear whether the proposed method can be generalized to other tasks, such as few-shot incremental semantic segmentation.
2. What is the ratio hyper-parameter between the loss $L_{match}$ and $L_{Cont}$? Is it set to 1? Would it make a difference in performance when setting different values?

---

> ### Author Response · Authors · 2025-11-21
> **The responses to Reviewer CEGG 1/2**
>
> > **W1: The paper missed several important and highly related references and comparative results.**
>
> **A1:** We sincerely appreciate your valuable suggestions. We have already included these papers, and they will be presented in our camera-ready version. Detailed results are presented in **Table 1, Table 2, and Figure 4** of the revised manuscript. For your convenience, the table below reports the AHM results across three FSCIL benchmarks. Overall, ConCM achieved the best AHM metrics, demonstrating **improvements of 12.53%, 15.83%, and 1.70%**, respectively.
>
> - **Mini-ImageNet:**
> |Methods|Base|1|2|3|4|5|6|7|8|AHM|
> | :--- | :--- | :--- | :--- | :--- | :--- | :--- | :--- | :--- | :--- | :--- |
> |MetaNC[3]|79.05|30.33|30.55|38.25|29.44|21.34|28.85|41.92|42.09|32.85|
> |SAVC[2]|80.02|42.40|37.80|36.56|37.20|33.53|31.72|31.98|32.94|35.52|
> |CLOSER[4]|76.97|43.58|40.36|40.41|39.35|37.76|35.72|36.10|37.58|38.86|
> | Mamba-FSCIL[1]|84.93|58.01|53.33|49.98|48.98|44.06|40.14|41.73|41.72|47.25|
> |ConCM(Ours)|83.97|70.34|66.59|63.38|59.59|57.05|53.95|53.49|53.92|59.78|
>
> - **CIFAR100:**
> |Methods|Base|1|2|3|4|5|6|7|8|AHM|
> | :--- | :--- | :--- | :--- | :--- | :--- | :--- | :--- | :--- | :--- | :--- |
> |MetaNC[3]|79.17|31.13|25.75|31.60|22.87|18.34|19.26|28.40|27.04|25.55|
> |SAVC[2]|78.07|38.69|34.95|30.42|28.90|30.30|31.59|31.68|30.79|32.16|
> |CLOSER[4]|75.95|48.31|44.98|41.99|40.87|40.02|41.80|41.64|40.23|42.48|
> |Mamba-FSCIL[1]|82.80|42.18|47.29|45.10|42.86|43.05|42.14|42.37|40.77|43.22|
> |ConCM(Ours)|82.82|72.27|67.33|60.09|57.08|54.93|55.21|52.95|52.51|59.05|
>
> - **CUB200:**
> |Methods|Base|1|2|3|4|5|6|7|8|9|10|AHM|
> | :--- | :--- | :--- | :--- | :--- | :--- | :--- | :--- | :--- | :--- | :--- | :--- | :--- |
> |MetaNC[3]|78.84|68.90|58.11|47.36|49.38|46.59|47.46|46.73|44.88|45.68|45.03|50.01|
> |SAVC[2]|80.00|64.29|60.89|54.56|58.27|55.03|57.92|57.10|57.33|57.36|57.72|58.05|
> |CLOSER[4]|79.40|63.69|62.29|58.21|59.48|59.04|59.13|59.96|57.91|59.31|59.01|59.80|
> |Mamba-FSCIL[1]|80.90|76.94|65.33|58.07|57.31|56.47|57.99|58.33|57.26|58.36|58.95|60.50|
> |ConCM(Ours)|80.52|75.24|66.04|59.78|60.10|59.48|60.24|60.78|59.35|60.57|60.39|62.20|
>
> [1] Mamba-FSCIL: Dynamic Adaptation with Selective State Space Model for Few-Shot Class Incremental Learning.
>
> [2] Learning With Fantasy: Semantic-Aware Virtual Contrastive Constraint for Few-Shot Class-Incremental Learning.
>
> [3] Learning optimal inter-class margin adaptively for few-shot class-incremental learning via neural collapse-based meta-learning.
>
> [4] Towards Better Representation Learning for Few-Shot Class-Incremental Learning.
>
> ---
> > **W2: The novelty of MPC compared to ProtoComNet[1] needs further clarification.**
>
> **A2:**  Thanks for your constructive feedback. We will discuss the novelty of MPC from the following perspectives:
>
> - **Different core objectives:**
>
>    - ProtoComNet[1] focuses on static completion of few-shot prototypes and is not applicable to incremental processes.
>
>    - The purpose of ConCM's MPC module is to maintain semantic consistency during the incremental process and support subsequent structural optimization.
>
> - **Different prototype aggregation methods:**
>
>    - ProtoComNet's aggregation weight relies solely on concatenating MLP outputs, potentially leading to the erroneous incorporation of irrelevant attributes.
>
>    - MPC employs a cross-attention mechanism to dynamically calculate the correlation weights between attributes and categories based on semantic and visual associations. This scheme enables focusing on attribute information highly relevant to the current category.
>
> - **The performance improvements in ConCM are more significant:**
>
>    - We adapted the ProtoComNet method to be compatible with the ConCM framework and conducted comparative experiments under the same baseline. Results on the mini-ImageNet dataset demonstrate that **MPC achieves a 2.44% improvement in AHM performance compared to ProtoComNet**.
>   |Methods|1|2|3|4|5|6|7|8|AHM|
>   | :--- | :--- | :--- | :--- | :--- | :--- | :--- | :--- | :--- | :--- |
>   |ConCM(ProtoComNet[1])|68.90|63.52|60.62|56.49|53.71|51.36|52.77|51.39|57.34|
>   |ConCM(MPC)|70.34|66.59|63.38|59.59|57.05|53.95|53.49|53.92|59.78|
>
> [1] Prototype completion for few-shot learning.

---

> ### Author Response · Authors · 2025-11-21
> **The responses to Reviewer CEGG 2/2**
>
> > **W3: Can the proposed method be transferred or adapted to pre-trained ViT models or CLIP models?**
>
> **A3:** Thanks for your question. ConCM adopts ResNet as its backbone in order to align with mainstream benchmarks. To evaluate its generalization, we conducted experiments on CIFAR-100 using a **ViT-B/16 backbone** (pre-trained on ImageNet-21k), which resulted in **a performance improvement of 2.67%** over OrCo. Therefore, the ConCM framework is not confined to specific backbone; its core lies in designing a universal framework for joint feature-structure optimization.
> |Methods|Base|1|2|3|4|5|6|7|8|AHM|
> | :--- | :--- | :--- | :--- | :--- | :--- | :--- | :--- | :--- | :--- | :--- |
> |OrCo-ViT|94.25|81.81|83.82|80.01|81.73|83.06|84.24|83.71|82.46|82.60|
> |ConCM-ViT|94.50|87.05|86.00|84.42|85.37|84.53|85.42|85.53|83.87|85.27|
>
> ---
> > **Q1: It is unclear whether the proposed method can be generalized to other tasks, such as few-shot incremental semantic segmentation.**
>
> **A4:** Thanks for your question. We agree that extending the proposed method to other tasks such as Incremental Few-Shot Semantic Segmentation (IFSS) is of great significance. However, this paper mainly focuses on the incremental recognition task of images. Due to the limited response time, we are sorry that we have not conducted more experimental validations yet. Nevertheless, we also considered the potential applicability of the ConCM framework to the IFSS task [1][2]: The MPC module theoretically can enhance pixel prototype modeling in segmentation tasks through attribute-level feature decoupling. However, a new challenge arises in addressing the mapping relationship between pixel-level features and higher-level attributes. The DSM module enhances representation generalization capabilities by optimizing feature space structure, which may prove beneficial for prototype-based IFSS. We will further elaborate on these points in the final version. Thank you once again for your valuable suggestions.
>
> [1] Prototype-based Incremental Few-Shot Semantic Segmentation.
>
> [2] Prototype-based Semantic Segmentation.
>
> ---
> > **Q2:  Loss ratio configuration and its impact on performance.**
>
> **A5:** We sincerely apologize for the lack of clarity in the manuscript. Throughout all experiments, **the loss ratio was consistently set to 1**.  Additionally, we have supplemented the impact of the loss hyperparameter  ${L_{{\text{Match}}}} + \eta {L_{Cont}}$ on performance in the mini-ImageNet, as shown below. Notably, our proposed method can consistently **maintain SOTA performance across all settings**.
> | $\eta$|0.5|0.75|1.00|1.25|1.5|
> | :--- | :--- | :--- | :--- |:--- | :--- |
> |AHM|58.97|59.06|59.78|59.38|58.73|

---

> > ### Comment · Reviewer_CEGG · 2025-11-27
> >
> > Thank you for the additional experiments and further clarification. However, I find that the newly added comparison results are all lower than the numbers reported in the original paper. For example, on the CUB dataset, SAVC reports 62.50 in the last session, and Mamba-FSCI reports 61.65. I understand that performance may vary slightly across different machines, but the gap from the original paper is quite large. I would appreciate the authors’ explanation and clarification.

---

> ### Author Response · Authors · 2025-11-28
> **Further responses to Reviewer CEGG**
>
> We sincerely apologize for the confusion caused. In the aforementioned comparative experiments, we report **the harmonic accuracy for each session** rather than accuracy, i.e. $HM_{t} = \frac{2 \times BAcc_{t} \times NAcc_{t}}{BAcc_{t} + NAcc_{t}}$, which is regarded as a balanced metric [a][b][c]. Where, $BAc{c_t}$ and $NAc{c_t}$ denote the base class accuracy and novel class accuracy for the $t$-th session, respectively. Additionally, due to our oversight, the **last session accuracy (FA)** of the comparison method is only **reported in the updated Table 1, Table 2, and Table 10**. Therefore, we provide a more detailed results as follows. In our reproduction, on the CUB200 dataset, **SAVC's last session accuracy is 60.82**, and **Mamba-FSCIL's last session accuracy is 61.67**. Our reproduction results are largely consistent with the original results. ConCM achieves stable performance on both AHM and FA metrics, whereas prior work suffers from degraded AHM performance due to excessive focus on the base class.
>
> - **mini-ImageNet:**
> |Methods|Base|1|2|3|4|5|6|7|8|AHM|FA|
> | :--- | :--- | :--- | :--- | :--- | :--- | :--- | :--- | :--- | :--- | :--- | :--- |
> |MetaNC[3]|79.05|30.33|30.55|38.25|29.44|21.34|28.85|41.92|42.09|32.85|53.11|
> |SAVC[2]|80.02|42.40|37.80|36.56|37.20|33.53|31.72|31.98|32.94|35.52|54.34|
> |CLOSER[4]|76.97|43.58|40.36|40.41|39.35|37.76|35.72|36.10|37.58|38.86|53.61|
> |Mamba-FSCIL[1]|84.93|58.01|53.33|49.98|48.98|44.06|40.14|41.73|41.72|47.25|59.36|
> |ConCM|83.97|70.34|66.59|63.38|59.59|57.05|53.95|53.49|53.92|59.78|59.92|
>
>
>
> - **CIFAR100:**
> |Methods|Base|1|2|3|4|5|6|7|8|AHM|FA|
> | :--- | :--- | :--- | :--- | :--- | :--- | :--- | :--- | :--- | :--- | :--- | :--- |
> |MetaNC[3]|79.17|31.13|25.75|31.60|22.87|18.34|19.26|28.40|27.04|25.55|51.99|
> |SAVC[2]|78.07|38.69|34.95|30.42|28.90|30.30|31.59|31.68|30.79|32.16|51.93|
> |CLOSER[4]|75.95|48.31|44.98|41.99|40.87|40.02|41.80|41.64|40.23|42.48|53.55|
> |Mamba-FSCIL[1]|82.80|42.18|47.29|45.10|42.86|43.05|42.14|42.37|40.77|43.22|57.51|
> |ConCM|82.82|72.27|67.33|60.09|57.08|54.93|55.21|52.95|52.51|59.05|58.33|
>
>
>
> - **CUB200:**
> |Methods|Base|1|2|3|4|5|6|7|8|9|10|AHM| FA|
> | :--- | :--- | :--- | :--- | :--- | :--- | :--- | :--- | :--- | :--- | :--- | :--- | :--- | :--- |
> |MetaNC[3]|78.84|68.90|58.11|47.36|49.38|46.59|47.46|46.73|44.88|45.68|45.03|50.01|50.62|
> |SAVC[2]|80.00|64.29|60.89|54.56|58.27|55.03|57.92|57.10|57.33|57.36|57.72|58.05|60.82|
> |CLOSER[4]|79.40|63.69|62.29|58.21|59.48|59.04|59.13|59.96|57.91|59.31|59.01|59.80|62.38|
> |Mamba-FSCIL[1]|80.90|76.94|65.33|58.07|57.31|56.47|57.99|58.33|57.26|58.36|58.95|60.50|61.67|
> |ConCM|80.52|75.24|66.04|59.78|60.10|59.48|60.24|60.78|59.35|60.57|60.39|62.20|62.66|
>
> Thank you once again for your valuable comments. If any issues remain, we would be grateful for your feedback. We truly appreciate your insights and the time you've dedicated to improving our work.
>
> [a] OrCo: Towards Better Generalization via Orthogonality and Contrast for Few-Shot Class-Incremental Learning
>
> [b] Few-Shot Class-Incremental Learning via Training-Free Prototype Calibration
>
> [c] Few-Shot Class-Incremental Learning from an Open-Set Perspective

---

### Official Review · Reviewer_R41s · 2025-11-05

**Soundness:** 3
**Presentation:** 2
**Contribution:** 3
**Rating:** 6
**Confidence:** 4

**Summary:**

This paper addresses the FSCIL problem by identifying two key challenges — feature inconsistency and structure inconsistency.
The proposed ConCM framework introduces (1) a Memory-aware Prototype Calibration (MPC) module that leverages semantic attributes to calibrate prototypes, and (2) a Dynamic Structure Matching (DSM) module that dynamically updates class geometry to maintain global consistency.
Experiments on multiple benchmarks show that ConCM achieves better stability and accuracy than prior FSCIL methods.

**Strengths:**

1. The proposed modules (MPC and DSM) are conceptually sound and complementary, leading to consistent and clear improvements across multiple benchmarks.

2. The paper provides clear theoretical motivation and connects the design of DSM with neural collapse geometry, giving the framework better interpretability.

**Weaknesses:**

1. The Memory-aware Prototype Calibration (MPC) relies on semantic attribute extraction from WordNet or class names, which might not generalize to datasets without clear textual labels.

2. Conceptually, ConCM extends previous geometry-based or orthogonality-driven FSCIL ideas (e.g., OrCo, NC-based approaches), so its originality mainly lies in how these components are unified.

3. Consider including at least one recent 2025 method to strengthen the comparison with up-to-date FSCIL approaches.

**Questions:**

How sensitive is the performance to the semantic quality of the extracted attributes in MPC? Would ConCM still work well if semantic information is noisy or unavailable?

---

> ### Author Response · Authors · 2025-11-21
> **The responses to Reviewer R41s 1/2**
>
> > **W1：The proposed method might not generalize to datasets without clear textual labels.**
>
> **A1:** Thanks for your thoughtful feedback. Extending the proposed method to datasets lacking clear textual labels is our primary objective. We hope the following clarifications will address your concerns:
> - **ConCM can still improve performance even without textual labels.** ConCM is a consistency-driven feature-structure joint optimization framework. If the dataset lacks clear textual labels, **removing MPC** (i.e., not optimizing features) and solely optimizing the embedding structure via DSM **still improves a 3.05% ~ 9.07% AHM improvement** compared to NC-FSCIL. The reason is that we constructed a more compatible dynamic feature embedding structure to continuously adapt to incremental tasks.
> |Methods|mini-ImageNet|CIFAR100|CUB200|
> | :--- | :--- | :--- | :--- |
> |NC-FSCIL|52.62|47.89|57.14|
> |ConCM w/o MPC|56.79|56.96|60.19|
>
> - **Improved approaches independent of text labels.** Within our framework, semantic attributes are not limited to extraction from textual labels. For instance, VLMs can be leveraged to construct self-supervised semantic knowledge bases, which allows for the direct discovery of implicit semantic attributes from images. The related work ATPrompt[1] has validated the feasibility.
>
> [1] Advancing Textual Prompt Learning with Anchored Attributes.
>
> ---
> > **W2: ConCM ’s originality mainly lies in how these components are unified.**
>
> **A2:**  Thanks for your comment. We would like to highlight some contributions of our work that may not have been sufficiently emphasized.
>
> - The current FSCIL ideas based on geometry or orthogonal driving only consider the static optimization of the structure. We explored the consistency-driven feature-structure joint dynamic optimization framework. ConCM is a more general framework that achieves the best performance.
>
> - Furthermore, we made adjustments to the MPC module, **making it compatible as a plug-and-play component** with the NC-FSCIL method. As a result, **AHM improved by ~3%**. Therefore, maintaining feature-structure consistency is a worthy approach to enhance the performance of the method. We aim to promote the further generalization of the FSCIL method by designing this complementary joint optimization framework.
> |Methods|Base|1|2|3|4|5|6|7|8|AHM|
> | :--- | :--- | :--- | :--- | :--- | :--- | :--- | :--- | :--- | :--- | :--- |
> |NC-FSCIL|84.07|62.34|61.04|55.93|53.13|49.68|47.08|46.22|45.57|52.62|
> |NC-FSCIL+MPC|83.93|68.87|62.83|59.15|54.36|52.51|48.76|47.87|49.95|55.53|

---

> ### Author Response · Authors · 2025-11-21
> **The responses to Reviewer R41s 2/2**
>
> > **W3: Consider including at least one recent 2025 method to strengthen the comparison with up-to-date FSCIL approaches.**
>
> **A3:**  Thanks for your valuable suggestion. We have supplemented the comparison with the 2025 approaches, including **ADBS**[1] and **FACL**[2]. Among the three benchmarks, the proposed method **achieved performance improvements of 14.20%, 26.07%, and 3.80% in AMH**. They over-optimizes base-class representations which disrupts cross-session balance and results in suboptimal AHM performance. In contrast, ConCM achieves more stable performance by maintaining dual consistency.
>
> - **mini-ImageNet:**
> |Methods|Base|1|2|3|4|5|6|7|8|AHM|
> | :--- |:---:|:---:|:---:|:---:|:---:|:---:|:---:|:---:|:---:|:---:|
> |ADBS[1]|79.53|44.88|45.60|39.00|42.61|41.68|40.58|42.93|43.12|42.58|
> |FACL[2]|86.23|50.21|40.35|37.45|39.59|35.61|32.32|32.66|34.22|37.80|
> |ConCM(Ours)|83.97|70.34|66.59|63.38|59.59|57.05|53.95|53.49|53.92|59.78|
>
> - **CIFAR100:**
> |Methods|Base|1|2|3|4|5|6|7|8|AHM|
> | :--- | :--- | :--- | :--- | :--- | :--- | :--- | :--- | :--- | :--- | :--- |
> |ADBS[1]|80.68|38.80|34.39|32.84|32.82|33.25|32.62|29.75|29.33|32.98|
> |FACL[2]|83.65|34.79|33.90|30.12|30.09|29.31|29.24|31.38|30.17|31.13|
> |ConCM(Ours)|82.82|72.27|67.33|60.09|57.08|54.93|55.21|52.95|52.51|59.05|
>
> - **CUB200:**
> |Methods|Base|1|2|3|4|5|6|7|8|9|10|AHM|
> | :--- | :--- | :--- | :--- | :--- | :--- | :--- | :--- | :--- | :--- | :--- | :--- | :--- |
> |ADBS[1]|79.45|66.40|61.02|54.46|59.29|55.59|57.34|58.10|57.69|57.09|56.96|58.40|
> |FACL[2]|80.88|68.33|61.21|57.25|57.38|57.26|58.54|59.76|58.11|56.99|58.22|59.30|
> |ConCM(Ours)|80.52|75.24|66.04|59.78|60.10|59.48|60.24|60.78|59.35|60.57|60.39|62.20|
>
> Additionally, we have **supplemented and compared with four highly relevant methods**. The results demonstrate that our method still achieves SOTA across all benchmarks. More detailed results are provided in **Table 1, Table 2 and Figure 4** of the revised manuscript.
>
> [1] Adaptive Decision Boundary for Few-Shot Class-Incremental Learning.
>
> [2] Strategic Base Representation Learning via Feature Augmentations for Few-Shot Class Incremental Learning.
>
> ---
>
> > **Q1 : Would ConCM still work well if semantic information is noisy or unavailable?**
>
> **A4:** We greatly appreciate this opportunity to clarify how ConCM handles semantic quality sensitivity and works when semantic information is unavailable.
> - **ConCM exhibits low sensitivity to semantic quality.** MPC has inherently considered the semantic sensitivity issue. The embedded cross-attention mechanism dynamically computes the distribution of attribute importance through semantic and visual associations. Optimized via meta-learning, this mechanism learns to assign high weights to discriminative attributes while suppressing the weights of noisy ones. In cross-domain settings (Table 15, Appendix G), **semantic differences lead to redundancy and noise** in the attribute pool. The following table further supplements the ablation analysis under these settings. **The MPC module achieves a 1.60% improvement** in AHM, demonstrating its continued effectiveness even in semantically noisy environments.
> |Methods|1|2|3|4|5|6|7|8|AHM|
> | :--- | :--- | :--- | :--- | :--- | :--- | :--- | :--- | :--- | :--- |
> |ConCM w/o MPC|68.37|56.56|49.22|46.10|44.90|45.69|43.33|43.23|49.68|
> |ConCM|69.56|57.10|49.55|47.02|47.85|48.16|45.62|45.36|51.28|
>
> - **ConCM maintains competitiveness when semantic information is unavailable.** When semantic information is completely unavailable, ConCM can independently leverage the DSM module to construct effective embedding structures. **As shown in A1**, focusing solely on the DSM module, our approach achieves **a performance improvement of 3.05%~9.07%** compared to prior work.

---

> ### Comment · Reviewer_R41s · 2025-11-25
>
> Thanks for your reply and additional results. However, my concern is how different qualities or misalignments in text names (like those from different VLMs or even randomly corrupted) affect performance in the MPC module. Simply saying that removing MPC still works well doesn’t convince me.

---

> ### Author Response · Authors · 2025-11-25
> **Further responses to Reviewer R41s**
>
> Thanks for your valuable time and response. To further response your concerns, we have supplemented the comparative analysis on semantic noise. Experimental results indicate that **semantic noise may cause a certain degree of performance degradation in the MPC module, but this is acceptable and reasonable**. When **50% noise** was added, the **AHM decreased by only 1.29%**. Even with significantly degraded semantic quality, **the MPC module still outperforms baseline methods**. Specifically,
> -  Firstly, please allow us to restate the MPC module’s process. The MPC extracts generalizable attributes from base classes to form a shared attribute pool. Then, it calibrates prototypes for novel categories by aggregating relevant encoding information from this pool through a cross-attention mechanism within the latent space.
>
> - Furthermore, for supplementary experiments on the mini-ImageNet benchmark, we randomly corrupted the text attribute pool at varying degrees. This process replaced attributes with meaningless text placeholders, simulating incomplete attribute extraction due to diminished text name quality.
>
> - Finally, the detailed results for different text corruption degrees are as follows. The results indicate that ConCM exhibits low semantic sensitivity and adapts well to noisy scenes. The reason is that meta-learning optimized MPC network suppresses the expression of irrelevant attributes through its embedded attention mechanism.
> |Methods|1|2|3|4|5|6|7|8|AHM|
> | :--- | :--- | :--- | :--- | :--- | :--- | :--- | :--- | :--- | :--- |
> |NC-FSCIL|62.34|61.04|55.93|53.13|49.68|47.08|46.22|45.57|52.62|
> |ConCM w/o MPC|68.77|63.06|59.09|56.26|53.76|50.45|51.32|51.63|56.79|
> |ConCM(Corruption: 75%)|68.64|65.76|60.65|57.98|55.06|51.93|51.54|51.98|57.94|
> |ConCM(Corruption: 50%)|68.98|66.74|61.23|57.82|56.06|52.21|52.14|52.79|58.49|
> |ConCM(Corruption: 25%)|69.52|66.46|62.03|58.64|56.89|53.15|52.96|53.11|59.10|
> |ConCM(Corruption: 0%)|70.34|66.59|63.38|59.59|57.05|53.95|53.49|53.92|59.78|
>
>
> We will include more detailed experimental validation and results analysis regarding semantic quality in the final version. Meanwhile, we hope that our response has addressed your concerns and alleviated doubts you may have had. If you have any further concerns, we would be grateful to hear your thoughts.

---

> ### Comment · Reviewer_R41s · 2025-11-25
>
> Thanks for the additional results, my concerns have addressed. I have updated my rating correspondingly

---

> > ### Author Response · Authors · 2025-11-25
> > **Thanks for increasing to 8!**
> >
> > Thank you so much for your effort engaged in the review phase and for increasing to 8! We’ll incorporate the discussion into the final version. Thanks again!

---

### Official Review · Reviewer_Hr2g · 2025-11-06

**Soundness:** 3
**Presentation:** 3
**Contribution:** 2
**Rating:** 6
**Confidence:** 4

**Summary:**

This paper introduces ConCM, a novel two-stage framework for few-shot class-incremental learning that explicitly addresses the “dual-consistency” dilemma. By emulating hippocampal associative memory, the MPC module first calibrates few-shot prototypes with semantically related attributes extracted from base classes; the DSM module then dynamically updates the embedding geometry to satisfy both equi-distant separation and maximal matching with the previous structure.

**Strengths:**

1.	The illustration is clear.
2.	The method and the theoretical analysis seem solid.
3.	The reported performance improvement is considerable.

**Weaknesses:**

1.	The proposed ConCM uses WordNet to extract semantic attributes from class names, to calibrate the prototypes of the new classes. It works fine in the standard benchmarks such as cifar, imagenet, but if there are no semantic label names for each class, how would such calibration work?
2.	Lack of results on benchmarks with more classes. ConCM relies on explicitly calibrating the feature space for each class. The paper does not discuss the influence of the increased number of classes, especially when the benchmarks in this paper contain at most 200 classes.
3.	Typo: Caption of Table 4 “mini-imagenet”.

**Questions:**

See weaknesses

---

> ### Author Response · Authors · 2025-11-21
> **The responses to Reviewer Hr2g**
>
> > **W1: If there are no semantic label names for each class, how would such calibration work?**
>
> **A1:** Thanks for your constructive question. We hope this clarification from the following two perspectives will be helpful:
> - **ConCM remains effective without semantic label names.** It addresses the dual consistency issue by innovatively proposing feature calibration (i.e., MPC) and structural matching (i.e., DSM). Therefore, even in the case where calibration is impossible, the DSM module still remains functional, **improving AHM by 3.05%-9.07%** compared to the baseline NC-FSCIL. It ensures generalization by constructing a more compatible embedding structure.
> |Methods|mini-ImageNet|CIFAR100|CUB200|
> | :--- | :---:| :---: | :---: |
> |NC-FSCIL|52.62|47.89|57.14|
> |ConCM w/o MPC|56.79|56.96|60.19|
>
>
> - **Attribute sources are not limited to labels.** Our core contribution is a universal framework for joint feature and structure optimization. Within this framework, attribute sources are not limited to semantic labels. One approach involves using VLMs as a self-supervised semantic knowledge base to directly extend latent attributes from images. ATPrompt [1] has partially explored this process. Therefore, establishing category-attribute mappings without requiring class names is our future research direction.
>
> [1] Advancing Textual Prompt Learning with Anchored Attributes.
>
> ---
> > **W2: Lack of results on benchmarks with more classes.**
>
> **A2:** Thanks for your suggestion. We supplemented the **ImageNet-1k benchmark** and **achieved a 9.3% improvement** on AHM compared to baseline methods (e.g., OrCo, NC-FSCIL). The number of categories in the ImageNet-1k benchmark has significantly increased (1,000 classes), which includes 600 base classes and is set up in 8 incremental sessions with a 50-way 5-shot configuration. When the number of categories increases significantly, the structural rigidity of baseline methods exacerbates feature confusion within narrow static spaces. In contrast, ConCM effectively mitigates this issue by constructing precise prototype anchors and dynamically optimizing geometric structures, delivering superior performance.
> |Methods|Base|1|2|3|4|5|6|7|8|AHM|
> | :--- | :--- | :--- | :--- | :--- | :--- | :--- | :--- | :--- | :--- | :--- |
> |OrCo|74.47|53.26|41.65|39.99|41.22|40.23|39.32|38.91|37.66|41.55|
> |NC-FSCIL|74.70|50.79|47.02|43.01|41.45|40.53|39.57|38.43|37.31|42.26|
> |ConCM(Ours)|75.25|66.04|58.69|54.46|51.11|49.10|46.33|43.95|42.87|51.56|
>
> ---
> > **W3: Typo: Caption of Table 4 “mini-imagenet”.**
>
> **A3:** Thanks for your careful review. The typo has been corrected. And we have also thoroughly proofread the manuscript to ensure no similar issues remain.

---

### Author Response · Authors · 2025-12-02
**Author Final Summary**

We sincerely appreciate the valuable time and insightful comments contributed by all reviewers. We also extend gratitude to AC for the additional effort and responsibility undertaken during this unexpected situation.

We propose a consistency-driven feature-structure joint optimization framework, i.e., **ConCM**, which achieves SOTA performance across multiple FSCIL benchmarks. During the rebuttal period, we improved our score from **an initial 6664 to a final 6866 (Hr2g: 6, R41s: 6 → 8, CEGG: 6, Tr9T: 4 → 6).** Three-quarters of the reviewers participated in the rebuttal, and all score improvements occurred prior to the information leak. We hereby promise that no ICLR code of conduct has been violated, and the entire rebuttal process strictly adhered to the double-blind review protocols. It is our sincere hope that the AC and PC will support the adjustment of scores.

The incident statement indicates the vulnerability was first reported at **UTC: 27 Nov, 2025, 15:09**. Therefore, we have summarized the timeline of this rebuttal in UTC time, hoping it will assist in AC’s decision-making.
1. We deeply regret that, due to the unexpected termination of the rebuttal, we were unable to engage in a thorough discussion with Reviewer Hr2g.
2. **Reviewer R41** indicated at **UTC: 25 Nov, 2025, 13:51** that concerns have been addressed, and **raised the rating from 6 to 8**.
3. Following **Reviewer CEGG's** initial response at **UTC: 27 Nov 2025, 22:34**, we replied again three hours later. Subsequent rebuttals were unexpectedly terminated.
4. **Reviewer Tr9T** stated at **UTC: 26 Nov 2025, 13:28**, “I'm generally satisfied with the responses,” and **raised the rating from 4 to 6**.

Here, we have highlighted the strengths mentioned in the reviewers' reviews.
1. **Clear motivation:** The paper is well written, and the motivation is clear (**CEGG**). The issue of feature distribution deviation and embedding structure adaptability is a key challenge in FSCIL, and this has been recognized by the community (**Tr9T**).
2. **The module design is rational and theoretically sound:** All the reviewers unanimously agreed that the proposed method is reasonable and reliable. In particular, **R41s** and **CEGG** note that this paper provides a clear theoretical basis, linking dynamic structural matching with neural collapse geometry to offer interpretability.
3. **Comprehensive experiment:** Similarly, all reviewers acknowledged the significant improvement in performance across multiple benchmarks (mini-ImageNet, CIFAR100, CUB200). **CEGG** highlighted that the ablation experiments were comprehensive and convincing.

We sincerely appreciate the reviewers' positive feedback. Meanwhile, we have carefully considered and responded to the issues they raised.
1. **Reviewer Hr2g** focused on the working mechanism when there are no semantic label names, and hoped to supplement the benchmark results that include more categories. Therefore, we clarified this working mechanism and conducted additional experiments. This included the ImageNet-1k benchmark, and the results showed that our method still SOTA performance.
2. **Reviewer R41s** encouraged us to include comparisons with more 2025 methods and to analyze the impact of semantic quality. We have accordingly refined our research and addressed the reviewer's concerns through thorough discussions. **As a result, the reviewer raised score from 6 to 8.**
3. **Reviewer CEGG** requested additional baseline comparisons and validation of generalization across other backbones and tasks. To this end, we provided further explanations and more experiments to validate that the method can be stably generalized. We addressed the reviewer's concerns and clarified the misunderstanding caused by metric differences in our final response.
4. **Reviewer Tr9T** suggested that discussing the distinctions between the proposed method and related work (PA, NC-FSCIL) would be helpful, and also noted that the assumption of Gaussian distribution transferability requires further validation. After thorough discussion, **Tr9T stated, "I'm generally satisfied with the responses, and I'll update my rating to 6."**

Finally, we would like to express our sincere gratitude once again to all reviewers and AC for their dedicated efforts. Their contributions have been crucial in enhancing the quality of our paper.

---

### Meta-Review · Area_Chair_VNeK · 2025-12-28

**Summary:**

This submission received **consistently positive initial scores**, with three reviewers leaning accept (Reviewers Hr2g, R41s, CEGG: all 6) and one reviewer initially leaning marginal reject (Reviewer Tr9T: 4). During the rebuttal and discussion phase, two reviewers explicitly raised their scores (Reviewer R41s: 6 → 8; Reviewer Tr9T: 4 → 6) after additional experiments and clarifications, resulting in a final score profile that is clearly above the acceptance threshold.

Across reviews, there is broad agreement that the paper is well motivated, technically sound, and delivers strong empirical performance on standard FSCIL benchmarks. While reviewers raised concerns regarding semantic reliance, theoretical assumptions, and incremental novelty relative to prior geometry-based FSCIL methods, the rebuttal addressed these concerns in a thorough and convincing manner, supported by additional experiments (e.g., ImageNet-1k scaling, semantic-noise robustness, covariance correlation analysis). Based on the initial scores, score updates after rebuttal, and the resolved concerns, my judgment as AC is to **Accept**.

**Reviewer Concerns:**

### Concerns effectively addressed by the rebuttal

* **Reliance on semantic label names / WordNet attributes**:
  Several reviewers questioned whether the MPC module would generalize to settings without reliable semantic labels
  *(Reviewers Hr2g, R41s, Tr9T)*.
  The authors demonstrated that DSM alone (without MPC) consistently improves over NC-FSCIL, and further provided semantic corruption experiments showing limited degradation even with heavily noisy attributes. Reviewer R41s explicitly confirmed that these concerns were addressed and raised the score to 8.

* **Scalability to larger numbers of classes**:
  Reviewer Hr2g raised concerns that benchmarks used in the paper contain relatively few classes (≤200). The authors added ImageNet-1k FSCIL experiments, showing substantial gains over strong baselines, which directly addressed scalability concerns.

* **Novelty relative to NC-FSCIL and PA-style methods**:
  Reviewer Tr9T questioned whether the proposed DSM is fundamentally different from NC-FSCIL and whether MPC is closely related to PA-style prototype transfer. The rebuttal provided a clear conceptual distinction (dynamic joint feature–structure optimization vs. static structure enforcement) and controlled comparisons, after which Reviewer Tr9T indicated satisfaction and updated the rating to 6.

* **Gaussian distribution and covariance transfer assumptions**:
  Reviewer Tr9T requested stronger justification for assuming covariance similarity across classes. The authors added empirical correlation analyses demonstrating a strong positive relationship between prototype similarity and covariance similarity, which the reviewer acknowledged as convincing.

* **Missing baselines and metric discrepancies**:
  Reviewer CEGG pointed out missing comparisons and apparent inconsistencies with reported numbers. The authors added multiple missing baselines (including recent methods) and clarified the distinction between AHM vs. FA metrics, resolving the confusion.

---

### Minor concerns that remain (non-blocking)

* **Dependence on external semantic knowledge for MPC**:
  Although robustness to noise and label absence was demonstrated, the strongest gains still occur when meaningful semantic attributes are available. The applicability of MPC in non-visual-semantic domains remains an open question
  *(Reviewers Hr2g, R41s)*.

* **Limited task scope**:
  The work focuses on FSCIL classification; extensions to tasks such as incremental segmentation are discussed but not validated experimentally
  *(Reviewer CEGG)*.

**Reviewer Scores:**

* **Reviewer Hr2g (initial: 6)** → **Likely unchanged**
  Positive on technical soundness and empirical gains; concerns on semantic reliance and scale were addressed.

* **Reviewer R41s (initial: 6)** → **Updated to 8**
  Explicitly confirmed that semantic sensitivity and comparison concerns were resolved after rebuttal.

* **Reviewer CEGG (initial: 6)** → **Likely unchanged**
  Strongly positive on motivation, theory, and experiments; remaining comments focused on completeness and clarity.

* **Reviewer Tr9T (initial: 4)** → **Updated to 6**
  Initially skeptical about novelty and Gaussian assumptions; additional analysis and experiments addressed the concerns.

---

### Decision · Program_Chairs · 2026-01-26

Accept (Poster)